# Medication adherence and its associated factors among oral pre-exposure prophylaxis (PrEP) users in China: The Real-world E-consumer Cohort of PrEP study

Qingyu Li[1⊙], Jingtao Zhou[1⊙], Yutong Xu[1], Huachun Zou[2], Chunqing Lin[3], Min Zhang[1], Jiayin Zheng[1], Yuhang Zhang[1], Siwen Huang[1], Zhiyi Zhao[1], Chi Ruan[1], Jiaqi Cheng[1], Jie Xu[4], Houlin Tang[4], Hui Xue[5], Sitong Luo[1,6]*

**1** Vanke School of Public Health, Tsinghua University, Beijing, China, **2** School of Public Health, Fudan University, Shanghai, China, **3** Department of Psychiatry and Biobehavioral Sciences, University of California, Los Angeles, California, United States of America, **4** National Center for AIDS/STD Control and Prevention, Chinese Center for Disease Control and Prevention, Beijing, China, **5** BluedHealth, Beijing, China, **6** Institute for Healthy China, Tsinghua University, Beijing, China

⊙ These authors contributed equally to this work.
* sitongluo@mail.tsinghua.edu.cn

## Abstract

### Background

Medication adherence is the key to success of HIV pre-exposure prophylaxis (PrEP). In China, the majority of real-world users purchase PrEP through e-commerce platforms, yet their adherence remains unaddressed. This cohort study aimed to evaluate PrEP adherence and its associated factors among e-consumers in China.

### Methods and findings

Eligible participants who had purchased PrEP online in the past 3 months were enrolled in the Real-world E-consumer Cohort of PrEP (RECOPE) in December 2023. Anonymous self-administered e-questionnaires were used at baseline and 1-, 3-, and 6-month follow-ups to collect data on PrEP adherence and potentially associated factors. Optimal adherence was defined as full compliance with the '2-1-1' dosing protocol for event-driven (ED) users and no missed pills in the past month for daily users. Generalized linear mixed-effects models and logistic regression were fitted to identify prognostic factors of PrEP adherence. Of 877 individuals invited, 680 were eligible and 657 completed the baseline survey (response rate 96.6%). The follow-up response rate at the 1-, 3-, and 6-month timepoints was 90.1% (592/657), 82.6% (543/657), and 80.1% (526/657), respectively. Among the 621 participants who had initiated PrEP at baseline, 529 (85.2%) used the ED regimen, and 92 (14.8%) used the daily regimen. Among ED users, the prevalence of optimal adherence at the four timepoints was 41.3% (154/373), 43.1% (146/339), 45.3% (139/307), and 52.6%

which permits unrestricted use, distribution, and reproduction in any medium, provided the original author and source are credited.

**Data availability statement:** All relevant data are within the manuscript and its Supporting information files.

**Funding:** This work was supported by the Capital's Funds for Health Improvement and Research (2024-2G-4402 to S.L.) https://wjw. beijing.gov.cn; the National Natural Science Foundation of China (82404326 to S.L.) https:// www.nsfc.gov.cn; the Beijing Nova Program (20250484872 to S.L.) https://kw.beijing. gov.cn; and the Prevention and Control of Emerging and Major Infectious Diseases – National Science and Technology Major Project (2025ZD01900800 to S.L.) https:// service.most.gov.cn. The funders had no role in study design, data collection and analysis, data interpretation, the decision to publish, or preparation of the manuscript.

**Competing interests:** I have read the journal's policy and the authors of this manuscript have the following competing interests: One of the authors, H.X. is employed by BluedHealth, the e-commerce platform for PrEP sales in China which assisted in the participant recruitment in the study. BluedHealth had no role in study design, data analysis and interpretation, manuscript writing, or the decision to submit the manuscript for publication. The other authors declare that they have no competing interests.

**Abbreviations:** aOR, adjusted odds ratio; CI, confidence interval; ED, event-driven; GLMM, generalized linear mixed model; MSM, men who have sex with men; PrEP, pre-exposure prophylaxis; RECOPE, Real-world E-consumer Cohort of PrEP; SD, standard deviation; WHO, World Health Organization.

(160/304), respectively. Among daily users, the prevalence of optimal adherence was 83.7% (77/92), 81.4% (83/102), 87.8% (86/98), 84.3% (75/89) at each survey wave, respectively. Within this group, the mean adherence rates in the past month ranged from 97.1% to 98.7% across waves. Among ED users, older age, receptive or versatile sexual role, and higher self-efficacy were positively associated with optimal adherence, while multiple same-sex partners, chemsex, and PrEP-related stigma were negatively associated factors. Selection bias, recall bias, social desirability bias, confounding bias, and limited representativeness were the main limitations of the study.

## Conclusion

The study found PrEP adherence was low among ED PrEP users who account for the majority of real-world e-consumers in China. Targeted interventions are suggested to prioritize enhancing users' understanding of medication instructions, promoting self-efficacy of maintaining adherence, and alleviating PrEP-related stigma. Additional attention should be given to ED users who have chemsex, insertive anal sex, multiple sexual partners, or a younger age.

---

## Author summary

### Why was this study done?

- Oral HIV pre-exposure prophylaxis (PrEP) is highly effective in preventing HIV acquisition only when users maintain optimal medication adherence.

- Currently, the majority of real-world PrEP users in China buy the medication online, yet little is known about their medication adherence.

- The cohort study aimed to evaluate the level and factors associated with PrEP adherence among real-world e-consumers in China.

### What did the researchers do and find?

- We recruited 657 e-consumers of PrEP from the largest online platform for PrEP services in China and collected their baseline, 1-, 3-, and 6-month follow-up survey data.

- We found that PrEP adherence was low among event-driven (ED) PrEP users who accounted for the majority of real-world e-consumers in China, while the adherence was high among daily PrEP users.

- Age, sexual role, number of same-sex partners, chemsex, self-efficacy of maintaining PrEP adherence, and PrEP-related stigma were significantly associated with PrEP adherence among the ED users.

## What do these findings mean?

- There is an urgent need to improve medication adherence among e-consumers of ED PrEP in China.

- Future targeted interventions can incorporate components of enhancing users' understanding of medication instructions, promoting self-efficacy of maintaining adherence, and alleviating PrEP-related stigma.

- Selection bias, recall bias, social desirability bias, confounding bias, and limited representativeness were the main limitations of the study.

## Introduction

HIV/AIDS continues to pose a significant public health threat around the world. In 2023, approximately 39.9 million people were living with HIV with 1.5 million new infections worldwide [1]. In China, the country reported over 1.3 million people living with HIV and 102,000 new diagnoses in 2024. Key populations, particularly men who have sex with men (MSM), continue to be disproportionately affected by the epidemic in the country [2].

Oral pre-exposure prophylaxis (PrEP), a highly effective biomedical prevention tool of HIV, can reduce the risk of HIV acquisition through sexual contact by up to 99% when taken as prescribed [3]. The World Health Organization (WHO) strongly endorses the use of oral PrEP for preventing HIV in key populations, such as MSM and sex workers [4]. In China, oral PrEP was officially approved in 2020, accompanied by the release of an Expert Consensus recommending its use among high-risk groups [5,6]. Currently, two primary PrEP regimens, daily and event-driven (ED) PrEP (also known as '2-1-1' regimen), are available in the country.

The success of PrEP is highly dependent on medication adherence in both clinical/implementation trials and real-world settings. A meta-analysis suggests that every 10% decline in PrEP adherence may reduce its effectiveness by approximately 13% [7]. Prior research reported various adherence rates across different contexts and populations, ranging from 10% to 84% [8]. In China, a retrospective study showed a poor adherence rate as low as 50% in an on-site PrEP promotion trial [9].

Recently, telemedicine approaches for PrEP delivery (e.g., purchasing PrEP via e-health commerce platforms) have been emerging in some countries (e.g., the United States, China, and Brazil) due to its nature of convenience and privacy protection, and whose uptake was accelerated by the COVID-19 pandemic [10,11]. In common e-consumer models of PrEP, users seeking to access PrEP through online platforms are required to complete an online medical history form and undergo a remote evaluation of eligibility by a professional provider following the PrEP guidelines. If approved, a certain number of pills (e.g., 90-day supply for daily PrEP users) are mailed directly to the clients, along with renewal instructions [10].

Currently in China, e-health commerce platforms are the primary channel for purchasing PrEP. According to the China CDC, nearly 70% of real-world PrEP users purchase PrEP online. The largest e-commerce platform is *Bluedhealth* operated by *BlueCity Group*, the most popular MSM-oriented community-based organization in Asia. Despite its advantages in convenience and privacy protection, the current online services of PrEP present some disadvantages in providing comprehensive instructions and counseling on PrEP use, which may pose distinct challenges for maintaining optimal medication adherence. For instance, when individuals seek to purchase PrEP through *BluedHealth*, the platform first requires them to complete an online screening questionnaire for HIV exposure risk and PrEP eligibility (e.g., recent HIV testing result, contraindications to PrEP) [12]. A physician then reviews the screening questions remotely and issues an electronic prescription, along with a standard written instructions on PrEP use, if PrEP eligibility criteria are met. No other interactive counseling are provided at this stage. If a client had questions, he/she must join another online counseling group on *BluedHealth* for further communication. In contrast, during offline clinic visits for PrEP acquisition, in-person, face-to-face, and one-on-one counseling can be directly provided by clinicians during and after the visit. This format allows

for immediate feedback and counseling adjustments, more nuanced history-taking of PrEP users, and direct observation of nonverbal cues. Therefore, compared with offline counseling, online counseling has a more complex procedure and less flexibility of interacting with doctors, which may negatively affect clients' understanding about PrEP use and ultimately reduce medication adherence.

Existing evidence on PrEP adherence is mostly from clinical/implementation/promotional trials or from real-world users who obtained PrEP from traditional offline channels [13–15]. Although the long-acting injectable PrEP has been approved in a growing number of countries and regions, oral PrEP is likely to remain a major option for the foreseeable future because of its significantly lower price and greater accessibility [16]. Thus, understanding and supporting oral PrEP adherence among individuals accessing PrEP through online platforms is critically important. This study aimed to evaluate PrEP medication adherence and examine the associated factors based on the prospective cohort of real-world online PrEP consumers.

## Methods

### Study design

The study established the Real-world E-consumer Cohort of PrEP (RECOPE) in China. It was a prospective cohort of PrEP e-consumers, designed to monitor PrEP adherence and explore associated risk factors. The participant recruitment and baseline assessment of the cohort were conducted from December 6, 2023 to January 4, 2024, and three follow-up assessments were completed at 1-, 3-, and 6-months. The study had obtained ethical approval from the Institutional Review Board of Tsinghua University (reference number: 20230195). This study is reported as per the STROBE guideline (S1 Checklist) and the CROSS Checklist (S2 Checklist).

### Participant recruitment

The participants were conveniently sampled and recruited through *BluedHealth*, the major online platform for PrEP services in China. *BluedHealth* provides HIV prevention and care services, such as offering sales of HIV preventive products (e.g., PrEP, post-exposure prophylaxis, condoms, HIV/syphilis/hepatitis B virus self-test kits) and related logistics and delivery. The average number of PrEP purchases per month on *BluedHealth* is over 4,000 [12]. During the study period, the price of PrEP on *BluedHealth* ranged from approximately 266 to 2,280 CNY (36.7–314.5 USD) per 30-tablet bottle, depending on the manufacturer, and was comparable to the in-store price in China.

The staff from *BluedHealth* were asked to send an internal invitation message via *BluedHealth* to their users who had purchased PrEP within the past one to three months. Users who expressed an interest in the study were initially screened for eligibility. The inclusion criteria were as follows: (1) being aged 18 years or older; (2) having purchased PrEP online and being qualified for PrEP use; (3) planning to use PrEP over the next six months; and (4) providing informed consent. The exclusion criteria included having communication barriers or reading disabilities reported by the recruiters/researchers.

Potentially eligible participants were then referred to the research team via WeChat, the most widely used messaging and social application in China (over 1.4 billion monthly active users, comparable to iMessage or WhatsApp), for further screening, informed consent, baseline survey, and follow-up surveys. WeChat was chosen because it is the most ubiquitous, secure, and familiar platform for the participants, which could help facilitate the communication between participants and investigators and increase the follow-up rates. The potential participants were clearly informed about the study objective, procedures, potential risks and benefits, and voluntary and anonymous nature of the study by the research staff. Those who were eligible and agreed to join the study were then asked to provide an oral informed consent by clicking the agreement button in an e-informed consent form sent by the researcher via WeChat to preserve anonymity. Each enrolled participant was assigned a unique ID for linking baseline and follow-up surveys.

## Baseline and follow-up data collection

The baseline and 1-, 3-, and 6-month follow-up surveys were conducted by completing an anonymous and self-administered e-questionnaire. The link to the e-questionnaires was sent through WeChat. The participants were asked to click the link and log into the survey platform (https://www.wjx.com) using their unique ID as the passcode. In each wave of the surveys, several reminders for completing the questionnaires were sent to the participants through WeChat before identifying a participant as non-response or loss to follow-up. Participants who did not respond or were loss to follow-up in a given survey wave were re-contacted and invited to participate in subsequent follow-up waves. Each survey took 5–10 min to complete. A *BluedHealth* voucher valued 50 CNY (approximately 7 USD) was provided upon completion of each valid survey. This voucher could be redeemed on *BluedHealth* for HIV prevention products and services, such as PrEP medications, condoms, and HIV/syphilis/HBV testing kits. To check questionnaire quality, three attention-check questions were embedded within each questionnaire. Participants who correctly answered at least two questions were included in the analytical sample.

## Measures

**Outcome variable: PrEP adherence.** Adherence to ED regimen was assessed by four items. First, the participants were asked whether they had engaged in sexual activity in the past one and three months, respectively (the 1-month follow-up survey only asked this question for the past one month). Those who answered "no" or reported having never initiated PrEP in a previous question were treated as missing data for PrEP adherence. Those who answered "yes" were then asked if they had taken PrEP for each sexual encounter in the corresponding time period (1 = every time, 2 = sometimes, and 3 = never). The participants who chose "every time" or "sometimes" were subsequently asked three adherence questions based on the '2-1-1' dosing guideline: (i) whether (yes or no) they took two pills of PrEP 2–24 hours before the sexual activity for each medication (time point A); (ii) whether they took one pill 24 hours after the first dose for each medication (time point B); and (iii) whether they took another pill 48 hours after the first dose for each medication (time point C). The participants who reported taking PrEP for each sexual encounter and chose "yes" to all three adherence questions were classified as "being adherent", while all others were classified as "non-adherent." Lastly, three binary outcome variables were constructed: past 1-month adherence based on baseline and all follow-up data, past 3-month adherence based on baseline and 3- and 6-month follow-up data, and consistent adherence based on 3- and 6-month follow-up data (defined as having optimal past 3-month adherence at both 3- and 6-month follow-up surveys).

Adherence to daily PrEP regimen was assessed with a single item: "In the past one month, did you have any occasions that you did not take PrEP on time or following the recommended dosage? (yes/no)". Participants who answered "no" were classified as "being adherent," and those who chose "yes" were categorized as "non- adherent" and subsequently asked to report how many PrEP pills they had missed in the past month. The adherence rate in the past month of each daily user was also calculated following the formula:

$$\text{Adherence rate } (\%) = \frac{30 - \text{number of missing pills}}{30} \times 100\%$$

In addition, all non-adherent participants were then asked to report reasons for missing doses. A list of reasons (e.g., I didn't know the correct way of taking PrEP; I forgot to take PrEP; I don't have access to PrEP in a timely manner; I experience adverse side effects) together with an open-ended option (Other, please specify) were provided.

**Exposure variables. Social-demographic characteristics:** The participants were asked about their age (in years), sex (male, female), ethnicity (Han, others), employment status (students, unemployed, employed), education level (Junior high school or below, senior high school or equivalent, bachelor degree or equivalent, master degree or above), marital status (married/living with a partner, unmarried/divorced/separated/widowed), and monthly income (≤3,000,

3,001–5,000, 5,001–7,000, 7,001–10,000, 10,001–15,000, 15,001–20,000, >20,000 CNY). Income was further recoded into a binary variable using 3,000 CNY (approximately 414 USD) as the threshold as it is the national average monthly income in China [17].

**Sexual behaviors:** The participants' sexual behaviors were measured, including whether having had chemsex (yes/no), number of sexual partners, commercial sexual behavior (yes/no), roles in sexual intercourse (insertive/receptive/versatile), and frequency of condoms use for each intercourse (never/occasionally/sometimes/often/always) in the corresponding time period. Frequency of condom use was further re-categorized as inconsistent condom use (never/occasionally/sometimes) and consistent condom use (often/always).

**PrEP-related variables:** Participants' PrEP knowledge on correct medication instructions was assessed via five questions for ED regimen (e.g., "Take 2 pills orally 2 to 24 hours before anticipated sexual activity; take 1 additional pill 24 hours after the initial dose following sexual activity; take 1 additional pill 48 hours after the initial dose following sexual activity") and six questions for daily users (e.g., "Take 1 pill every 24 hours; high-risk behavior for HIV should only be undertaken after taking the medication daily for 7 consecutive days"). Response options included "yes," "no," and "uncertain." Participants who selected all correct options were coded as "having correct knowledge".

PrEP-related stigma was measured by the 11-item Chinese version of HIV PrEP Stigma Scale (e.g., "Someone taking PrEP would be seen by others slutty) [18]. A summative score ranging from 11 to 55 ($\alpha$ in this study = 0.81) was calculated by summing item scores (1 = strongly disagree, 5 = strongly agree), with a higher score indicating greater stigma.

Self-efficacy of PrEP adherence was evaluated by six items adapted from the PrEP adherence self-efficacy scale developed by Walsh (e.g., "Even if my sexual partner is unwilling, I am confident in continuing PrEP use") [19,20]. A summative score ranging from 6 to 30 ($\alpha$ in this study = 0.83) was created by summing item scores (1 = strongly disagree, 5 = strongly agree), with a higher score indicating greater self-efficacy.

**Social-psychological variables:** Depressive symptoms were assessed by the Patient Health Questionnaire-9 (PHQ-9) [21], which has been validated in Chinese MSM populations and demonstrated strong psychometric properties [22]. The questionnaire asked the participants about their frequency of specific symptoms experienced over the past two weeks, with each item rated on a 4-point Likert scale ranging from 0 (not at all) to 3 (nearly every day). A summative score ranging from 0 to 27 was created by adding up the score of each item, and was categorized into standard severity groups for descriptive analysis: none/minimal (0−4), mild (5−9), moderate (10−14), moderately severe (15−19), and severe (20−27) ($\alpha$ in this study = 0.90). In the regression analyses, the continuous score of PHQ-9 was used to minimize information loss.

Psychological resilience was measured by the 2-item Connor-Davidson Resilience Scale (CD-RISC-2) [23], which has been validated among MSM populations [24]. The items included "Able to adapt to change" and "Tend to bounce back after illness or hardship". Responses were rated on a 5-point Likert scale ranging from 0 (never) to 4 (always). A summative score ranging from 0 to 8 was created by adding up the scores of the two items ($\alpha$ in this study = 0.84).

## Statistical analysis

First, descriptive analyses were performed to summarize the distributions of all the socio-demographic and psycho-behavioral variables across the study waves, both for the overall sample and sub-samples by PrEP regimens. The Chi-square test and $t$ test were conducted to examine differences in the variable distributions between study waves. The prevalence of optimal PrEP adherence at each wave was calculated for ED and daily users separately.

Second, regression analyses were performed to identify prognostic factors of PrEP adherence by examining statistical significance of crude and adjusted associations between the exposure variables (socio-demographic and psycho-behavioral measures) and the adherence outcomes. The prognosis factor strategy is useful for helping identify targets and enhancing design of future interventions on PrEP adherence [25]. For ED users, three regression models were fitted for the three adherence outcomes: (1) **model 1** included a univariate generalized linear mixed model (GLMM) with a logit link

PLOS Medicine

and random intercept to evaluate the statistical significance of crude association between each exposure variable and the past three-month adherence, as well as a multivariable GLMM including all exposure variables of interest in order to evaluate the significance of their independent associations with the outcome after controlling potential confounders (covariates); (2) **model 2** were similar univariate and multivariable GLMM for the past one-month adherence outcome; (3) **model 3** included a univariate and a multivariable logistic regression model for the consistent adherence outcome, in which baseline socio-demographics and averaged time-varying values at 3- and 6-months for other exposure variables were included as independent variables. Due to the small sample size (n = 96) of daily users, a univariable GLMM was fitted to examine the significance of the association between each exposure variable and the adherence rate in the past month.

All the above GLMM included fixed effects representing the average (population-level) association of the exposure variables with adherence outcomes, and random effects accounting for within-participant correlation of repeated measures across different survey waves. The models did not include interaction terms between time-points and the exposure variables, as it was assumed that the effects of exposure variables on PrEP adherence would remain relatively stable due to the limited number of repeated measurements in the study (four survey waves over six months) [26–28]. All the analyses were conducted on available data without imputation. Statistical significance was defined as a two-sided $p$-value < 0.05. The R software (version 4.4.2) was used to perform all the statistical analyses.

## Results

### Sample characteristics

Of the 877 individuals invited, 680 were eligible for the study and sent the baseline questionnaire (146 did not respond to the study invitation, 33 were not interested for participation after further communication with the recruiters, 18 had no plan to use PrEP in the future six months), and 657 completed the baseline survey (23 did not respond to the baseline questionnaire) and were finally enrolled in the RECOPE cohort. According to the American Association for Public Opinion Research (AAPOR)'s standard definitions, the response rate of the baseline survey was 96.6% (657/680). At the 1-, 3-, and 6-month follow-ups, 90.1% (592/657), 82.6% (543/657), and 80.1% (526/657) of the participants responded to the questionnaire surveys and were successfully followed-up, with 582, 539, and 520 being included in the analysis after excluding those failing the quality check of questionnaires (Fig 1). The patterns of loss to follow-up in the study were shown in S1 Table.

The distributions of socio-demographics in total and by regimens across different survey waves are presented in Table 1. For the total sample at baseline, the mean age was 29.9 years (standard deviation [SD] = 6.4). All the participants were assigned male sex at birth and reported to be MSM. Most participants were of Han ethnicity (n = 615, 93.6%), employed (n = 527, 80.2%), having a college education or above (n = 624, 95.0%), unmarried (n = 567, 86.3%), and having reported a monthly income over 7,000 CNY (n = 385, 58.6%). Of the 621 (94.5%) participants who had already initiated PrEP at baseline, 529 (85.2%) used the ED regimen and 92 (14.8%) took the daily regimen. There was no evidence of a difference (all $p > 0.05$) in the distributions of socio-demographic characteristics, either overall or by regimen, across the survey waves. The socio-demographic characteristics of those who had not initiated PrEP reported in each wave are shown in S2 Table.

### Distributions of exposure variables

Table 2 shows the distributions of sexual behaviors, PrEP-related variables, and psychosocial variables in total and by regimen types at each survey wave. In general, the majority of the total sample (78.4%–90.4%) reported recent sexual activities across all survey waves. Prevalence of high-risk sexual behaviors in the total sample were 38.9%−42.7% for chemsex, 37.7%–43.8% inconsistent condom use, 69.4%–81.0% for having multiple sexual partners, and 7.3%–9.4% for commercial sex. About 44.3%–48.5% of the total sample reported their sexual role as receptive or versatile. The majority of the participants (69.3%−76.2%) demonstrated correct knowledge of PrEP medication. The mean score of the

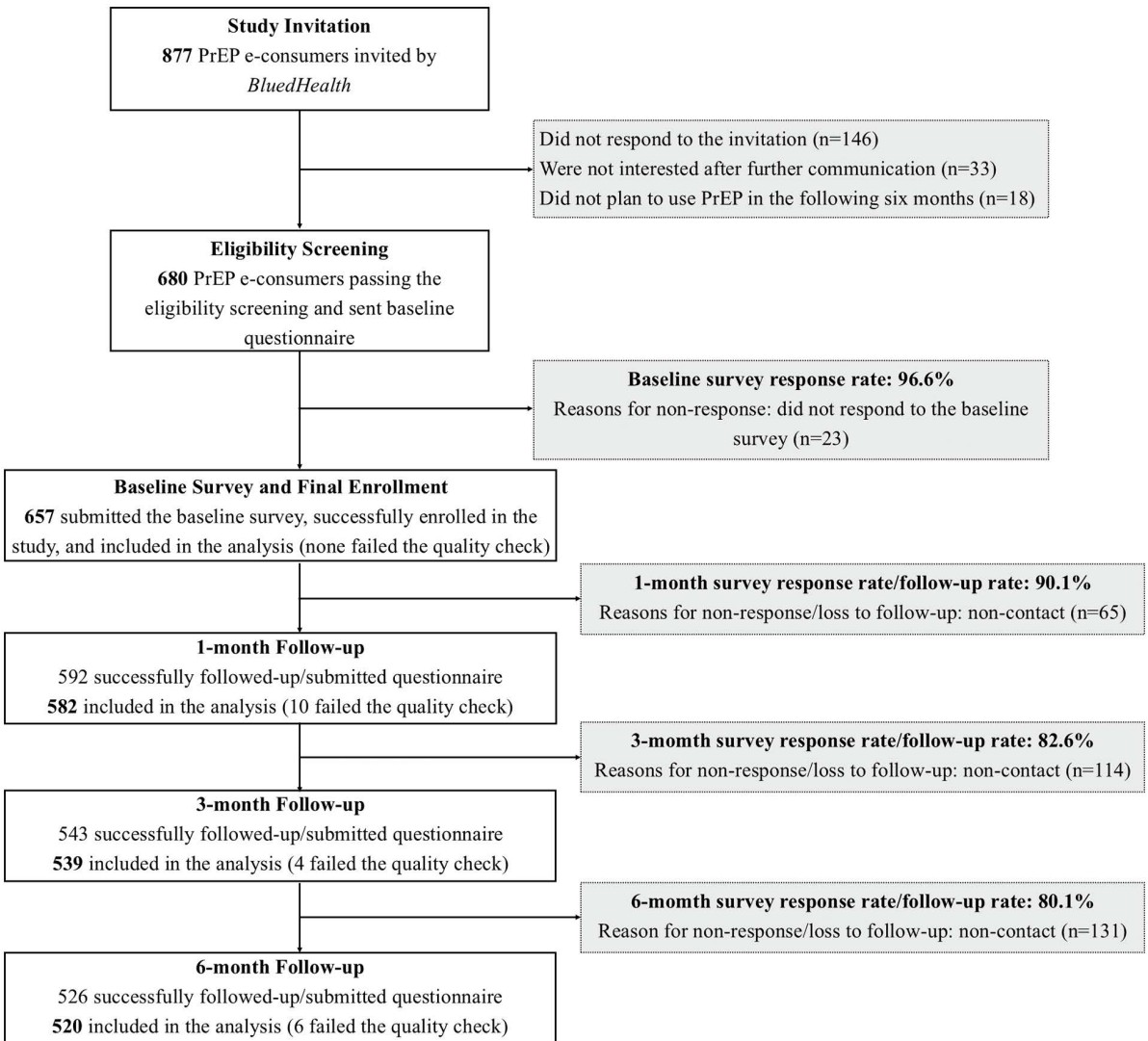

**Fig 1. Flowchart of the RECOPE study.** The figure shows the number of people who were invited, screened for eligibility, completed the baseline survey, finally enrolled, and successfully followed up, as well as the reasons for unenrollment and non-response/loss to follow-up. **Note**: Some participants completed surveys at different time points; detailed patterns of missing responses are provided in S1 Table.

self-efficacy scale ranged from 3.9 to 4.4. About 63.1%–70.2% reported mild to severe depressive symptoms. The mean score of the resilience scale ranged 5.2–5.8. There was statistical evidence ($p < 0.05$) of a difference in sexual activity status, number of sexual partners, PrEP knowledge, self-efficacy for PrEP adherence, and resilience scores across survey waves in the total sample and ED users. Among daily users, we also observed statistical evidence ($p < 0.05$) of differences across waves in self-efficacy for PrEP adherence, PrEP-related stigma, and resilience.

## PrEP adherence outcomes

The levels of the PrEP adherence outcomes across different waves are presented in Fig 2. Among the ED users, the prevalence of optimal adherence in the past one month was 41.3% (154/373), 43.1% (146/339), 45.3% (139/307), and

Table 1. Distributions of socio-demographic characteristics in the RECOPE study.

| Variables | Total | | | | $P^a$ | ED regimen | | | | $P^a$ | Daily regimen | | | | $P^a$ |
|---|---|---|---|---|---|---|---|---|---|---|---|---|---|---|---|
| | Baseline (N=657) | 1 month (N=582) | 3 month (N=539) | 6 month (N=520) | | Baseline (n=529) | 1 month (n=458) | 3 month (n=428) | 6 month (n=422) | | Baseline (n=92) | 1 month (n=102) | 3 month (n=98) | 6 month (n=88) | |
| **Age (years, Mean ± SD)** | 29.9±6.4 | 29.8±6.5 | 29.8±6.4 | 29.9±6.4 | 0.976 | 29.9±6.4 | 29.6±6.6 | 29.7±6.4 | 29.72±6.5 | 0.938 | 30.4±6.4 | 31.00±6.1 | 30.17±6.0 | 31.14±6.1 | 0.655 |
| 18–29 | 359 (54.6) | 322 (55.3) | 297 (55.1) | 278 (53.4) | | 289 (54.6) | 260 (56.8) | 240 (56.1) | 235 (55.7) | | 48 (52.2) | 48 (47.1) | 50 (51.0) | 38 (43.2) | |
| 30–39 | 246 (37.5) | 214 (36.8) | 202 (37.5) | 203 (39.1) | | 198 (37.5) | 164 (35.8) | 154 (36.0) | 155 (36.8) | | 37 (40.2) | 47 (46.1) | 43 (43.9) | 43 (48.9) | |
| >40 | 52 (7.9) | 46 (7.9) | 40 (7.4) | 39 (7.5) | | 42 (7.9) | 34 (7.4) | 34 (7.9) | 32 (7.6) | | 7 (7.6) | 7 (6.9) | 5 (5.1) | 7 (8.0) | |
| **Designated Sex at Birth** | | | | | N/A | | | | | N/A | | | | | N/A |
| Male | 657 (100.0) | 582 (100.0) | 539 (100.0) | 520 (100.0) | | 529 (100.0) | 458 (100.0) | 428 (100.0) | 422 (100.0) | | 92 (100.0) | 102 (100.0) | 98 (100.0) | 88 (100.0) | |
| Female | 0 (0.0) | 0 (0.0) | 0 (0.0) | 0 (0.0) | | 0 (0.0) | 0 (0.0) | 0 (0.0) | 0 (0.0) | | 0 (0.0) | 0 (0.0) | 0 (0.0) | 0 (0.0) | |
| **Ethnicity** | | | | | 0.963 | | | | | 0.977 | | | | | 0.965 |
| Han | 615 (93.6) | 547 (94.0) | 508 (94.2) | 491 (94.3) | | 497 (94.0) | 432 (94.3) | 405 (94.6) | 398 (94.3) | | 85 (92.4) | 94 (92.2) | 92 (93.9) | 82 (93.2) | |
| Others | 42 (6.4) | 35 (6.0) | 31 (5.8) | 29 (5.7) | | 32 (6.0) | 26 (5.7) | 23 (5.4) | 24 (5.7) | | 7 (7.6) | 8 (7.8) | 6 (6.1) | 6 (6.8) | |
| **Employment** | | | | | 0.978 | | | | | 0.926 | | | | | 0.540 |
| Students/Unemployed | 130 (19.8) | 119 (20.4) | 112 (20.8) | 106 (20.3) | | 114 (21.6) | 105 (22.9) | 96 (22.4) | 96 (22.4) | | 9 (9.8) | 10 (9.8) | 13 (13.3) | 6 (6.8) | |
| Employed | 527 (80.2) | 463 (79.6) | 427 (79.2) | 414 (79.7) | | 415 (78.4) | 353 (77.1) | 332 (77.6) | 333 (77.6) | | 83 (90.2) | 92 (90.2) | 85 (86.7) | 82 (93.2) | |
| **Education Level** | | | | | 0.999 | | | | | 1.000 | | | | | 0.952 |
| Junior high school or below | 4 (0.6) | 2 (0.3) | 3 (0.6) | 3 (0.6) | | 4 (0.8) | 2 (0.4) | 3 (0.7) | 3 (0.7) | | 0 (0.0) | 0 (0.0) | 0 (0.0) | 0 (0.0) | |
| Senior high school/Vocational high school/Technical secondary school | 29 (4.4) | 24 (4.1) | 22 (4.1) | 25 (48) | | 23 (4.3) | 18 (3.9) | 17 (4.0) | 18 (4.3) | | 6 (6.5) | 6 (5.9) | 5 (5.1) | 7 (8.0) | |
| College diploma (Associate degree)/Bachelor's degree | 436 (66.4) | 384 (66.0) | 358 (66.4) | 343 (65.9) | | 353 (66.7) | 309 (67.5) | 287 (67.1) | 282 (66.8) | | 54 (58.7) | 58 (56.9) | 61 (62.2) | 54 (61.4) | |
| Master's degree or above | 188 (28.6) | 172 (29.6) | 156 (28.9) | 149 (28.7) | | 149 (28.2) | 129 (28.2) | 121 (28.3) | 119 (28.2) | | 32 (34.8) | 38 (37.3) | 32 (32.7) | 27 (30.7) | |
| **Marital Status** | | | | | 0.646 | | | | | 0.524 | | | | | 0.898 |
| Married or living with a partner | 90 (13.7) | 82 (14.1) | 81 (15.0) | 84 (16.2) | | 77 (14.6) | 63 (13.8) | 68 (15.9) | 72 (17.1) | | 11 (12.0) | 16 (15.7) | 13 (13.3) | 12 (13.6) | |
| Unmarried/Divorced/Separated/Widowed | 567 (86.3) | 500 (85.9) | 458 (85.0) | 436 (83.8) | | 452 (85.4) | 395 (86.2) | 360 (84.1) | 350 (82.9) | | 81 (88.0) | 86 (84.3) | 85 (86.7) | 76 (86.4) | |

*(Continued)*

**Table 1.** (Continued)

| Variables | Total | | | | $P^a$ | ED regimen | | | | $P^a$ | Daily regimen | | | | $P^a$ |
|---|---|---|---|---|---|---|---|---|---|---|---|---|---|---|---|
| | Baseline (N=657) | 1 month (N=582) | 3 month (N=539) | 6 month (N=520) | | Baseline (n=529) | 1 month (n=458) | 3 month (n=428) | 6 month (n=422) | | Baseline (n=92) | 1 month (n=102) | 3 month (n=98) | 6 month (n=88) | |
| **Monthly income (CNY)**[b] | | | | | 1.000 | | | | | 1.000 | | | | | 0.961 |
| ≤3,000 | 77 (11.7) | 72 (12.4) | 67 (12.4) | 61 (11.8) | | 62 (11.7) | 61 (13.3) | 55 (12.9) | 54 (12.8) | | 9 (9.8) | 6 (5.9) | 9 (9.2) | 5 (5.7) | |
| 3,001–5,000 | 92 (14.0) | 82 (14.1) | 77 (14.3) | 75 (14.5) | | 78 (14.7) | 70 (15.3) | 64 (15.0) | 67 (15.9) | | 9 (9.8) | 11 (10.8) | 13 (13.3) | 8 (9.1) | |
| 5,001–7,000 | 103 (15.7) | 87 (14.9) | 82 (15.2) | 78 (15.0) | | 91 (17.2) | 73 (15.9) | 71 (16.6) | 68 (16.1) | | 9 (9.8) | 12 (11.8) | 11 (11.2) | 10 (11.4) | |
| 7,001–10,000 | 121 (18.4) | 114 (19.6) | 107 (19.9) | 105 (20.2) | | 95 (18.0) | 84 (18.3) | 77 (18.0) | 82 (19.4) | | 18 (19.6) | 23 (22.5) | 25 (25.5) | 20 (22.7) | |
| 10,001–15,000 | 127 (19.3) | 110 (18.9) | 104 (19.3) | 104 (19.8) | | 101 (19.1) | 89 (19.4) | 81 (18.9) | 80 (18.9) | | 19 (20.7) | 19 (18.6) | 21 (21.4) | 22 (25.0) | |
| 15,001–20,000 | 57 (8.7) | 48 (8.2) | 42 (7.8) | 35 (6.7) | | 38 (7.2) | 33 (7.2) | 29 (6.8) | 21 (5.0) | | 15 (16.3) | 13 (12.7) | 11 (11.2) | 12 (13.6) | |
| >20,000 | 80 (12.2) | 69 (11.9) | 60 (11.1) | 62 (11.9) | | 64 (12.1) | 48 (10.5) | 51 (11.9) | 50 (11.9) | | 13 (14. 1) | 18 (17.6) | 8 (8.2) | 11 (12.5) | |

[a]The Chi-squared test was used for categorical variables and $t$ test was used for continuous variables.

[b]1 CNY=0.14 USD.

SD, Standard deviation; ED, event-driven; CNY, Chinese Yuan.

**Table 2.** Distributions of sexual behaviors, PrEP-related variables, and psychosocial variables in the RECOPE study.

| Variables | Total | | | | P[a] | ED regimen | | | | P[a] | Daily regimen | | | | P[a] | Continuous Sexually active[b] |
|---|---|---|---|---|---|---|---|---|---|---|---|---|---|---|---|---|
| | Baseline (N=657) | 1 month (N=582) | 3 month (N=539) | 6 month (N=520) | | Baseline (n=529) | 1 month (n=458) | 3 month (n=428) | 6 month (n=422) | | Baseline (n=92) | 1 month (n=102) | 3 month (n=98) | 6 month (n=88) | | (n=310) |
| **Sexually active** | | | | | <0.001 | | | | | <0.001 | | | | | 0.476 | |
| Yes | 578 (88.0) | 454 (78.4) | 481 (89.2) | 470 (90.4) | | 458 (86.6) | 339 (74.0) | 377 (88.1) | 374 (88.8) | | 88 (95.7) | 100 (98.0) | 94 (95.9) | 88 (98.9) | | 310 (100.0) |
| No | 79 (12.0) | 128 (21.6) | 58 (10.8) | 50 (9.6) | | 71 (13.4) | 119 (26.0) | 51 (11.9) | 47 (11.2) | | 4 (4.3) | 2 (2.0) | 4 (4.1) | 1 (1.1) | | 0 (0.0) |
| **Chemsex[c]** | | | | | 0.564 | | | | | 0.338 | | | | | 0.803 | |
| Yes | 247 (42.7) | 182 (40.1) | 202 (42.0) | 183 (38.9) | | 202 (44.1) | 134 (39.5) | 157 (41.6) | 143 (38.2) | | 35 (39.8) | 44 (44.0) | 44 (46.8) | 37 (42.0) | | 160 (51.6) |
| No | 331 (57.3) | 272 (59.9) | 279 (58.0) | 287 (61.1) | | 256 (55.9) | 205 (60.5) | 220 (58.4) | 231 (61.8) | | 53 (60.2) | 56 (56.0) | 50 (53.2) | 51 (58.0) | | 150 (48.4) |
| **Condom use[c]** | | | | | 0.090 | | | | | 0.303 | | | | | 0.790 | |
| Inconsistent | 218 (37.7) | 201 (44.3) | 210 (43.7) | 206 (43.8) | | 164 (35.8) | 141 (41.6) | 154 (40.8) | 151 (40.4) | | 46 (52.3) | 56 (56.0) | 55 (58.5) | 52 (59.1) | | 144 (46.5) |
| Consistent | 360 (62.3) | 253 (55.7) | 271 (56.3) | 264 (56.2) | | 294 (64.2) | 198 (58.4) | 223 (59.2) | 223 (59.6) | | 42 (47.7) | 44 (44.0) | 39 (41.5) | 36 (40.9) | | 166 (53.5) |
| **Sexual partner[c]** | | | | | <0.001 | | | | | <0.001 | | | | | 0.240 | |
| 1 | 110 (19.0) | 139 (30.6) | 125 (26.0) | 132 (28.1) | | 84 (18.3) | 108 (31.9) | 109 (28.9) | 115 (30.7) | | 17 (19.3) | 25 (25.0) | 13 (13.8) | 15 (17.0) | | 53 (17.1) |
| >1 | 468 (81.0) | 315 (69.4) | 356 (74.0) | 338 (71.9) | | 374 (81.7) | 231 (68.1) | 268 (71.1) | 259 (69.3) | | 71 (80.7) | 75 (75.0) | 81 (86.2) | 73 (83.0) | | 257 (82.9) |
| **Sexual role[c]** | | | | | 0.649 | | | | | 0.650 | | | | | 0.600 | |
| Receptive or versatile | 267 (46.2) | 220 (48.5) | 222 (46.2) | 208 (44.3) | | 222 (48.5) | 172 (50.7) | 176 (46.7) | 174 (46.5) | | 34 (38.6) | 44 (44.0) | 44 (46.8) | 34 (38.6) | | 155 (50.0) |
| Insertive | 311 (53.8) | 234 (51.5) | 259 (53.8) | 262 (55.7) | | 236 (51.5) | 167 (49.3) | 201 (53.3) | 200 (53.5) | | 54 (61.4) | 56 (56.0) | 50 (53.2) | 54 (61.4) | | 155 (50.0) |
| **Commercial sex[c]** | | | | | 0.676 | | | | | 0.673 | | | | | 0.892 | |
| Yes | 50 (8.7) | 36 (7.9) | 35 (7.3) | 44 (9.4) | | 39 (8.5) | 27 (8.0) | 24 (6.4) | 31 (8.3) | | 9 (10.2) | 9 (9.0) | 10 (10.6) | 11 (12.5) | | 34 (11.0) |
| No | 528 (91.3) | 418 (92.1) | 446 (92.7) | 426 (90.6) | | 419 (91.5) | 312 (92.0) | 353 (93.6) | 343 (91.7) | | 79 (89.8) | 91 (91.0) | 84 (89.4) | 77 (87.5) | | 276 (89.0) |
| **PrEP knowledge[d]** | | | | | 0.014 | | | | | 0.002 | | | | | 0.590 | |
| Correct | 455 (69.3) | 416 (71.5) | 410 (76.1) | 396 (76.2) | | 390 (73.7) | 356 (77.7) | 356 (83.2) | 341 (81.0) | | 42 (45.7) | 49 (48.0) | 46 (46.9) | 49 (55.1) | | 233 (75.2) |
| In correct | 202 (30.7) | 166 (28.5) | 129 (23.9) | 124 (23.8) | | 139 (26.3) | 102 (22.3) | 72 (16.8) | 80 (19.0) | | 50 (54.3) | 53 (52.0) | 52 (53.1) | 40 (44.9) | | 77 (24.8) |
| **Self-efficacy of PrEP adherence** | 3.9±1.8 | 4.2±1.8 | 4.4±1.8 | 4.3±1.9 | <0.001 | 3.8±1.9 | 4.0±1.8 | 4.3±1.8 | 4.1±1.9 | <0.001 | 4.6±1.6 | 5.1±1.4 | 5.1±1.3 | 5.1±1.6 | 0.041 | 4.1±1.7 |
| **PrEP-related stigma** | 25.1±6.9 | – | 25.0±7.1 | 24.9±6.6 | 0.084 | 25.3±6.7 | – | 25.4±6.9 | 24.7±6.7 | 0.275 | 23.2±7.3 | – | 23.3±7.4 | 25.7±6.6 | 0.011 | 25.1±4.6 |
| **Depressive symptoms** | 7.2±5.3 | – | 7.4±5.3 | 7.2±5.2 | 0.267 | 7.1±5.3 | – | 7.3±5.3 | 7.0±5.0 | 0.283 | 7.4±5.4 | – | 7.5±5.7 | 8.0±5.7 | 0.477 | 7.2±3.7 |
| None | 215 (32.7) | – | 172 (31.9) | 155 (29.8) | | 175 (33.1) | – | 135 (25.3) | 129 (30.6) | | 29 (31.5) | – | 30 (32.6) | 23 (25.8) | | 65 (21.0) |
| Mild | 280 (42.6) | – | 233 (43.2) | 249 (47.9) | | 223 (42.2) | – | 196 (37.1) | 207 (49.2) | | 40 (43.5) | – | 27 (29.3) | 40 (44.9) | | 103 (49.4) |
| Moderate | 97 (14.8) | – | 73 (13.5) | 67 (12.9) | | 79 (14.9) | – | 61 (11.5) | 50 (11.9) | | 13 (14.1) | – | 9 (9.8) | 14 (15.7) | | 56 (18.1) |

*(Continued)*

**Table 2.** (Continued)

| Variables | Total | | | | $P^a$ | ED regimen | | | | $P^a$ | Daily regimen | | | | $P^a$ | Continuous Sexually active[b] |
|---|---|---|---|---|---|---|---|---|---|---|---|---|---|---|---|---|
| | Baseline (N=657) | 1 month (N=582) | 3 month (N=539) | 6 month (N=520) | | Baseline (n=529) | 1 month (n=458) | 3 month (n=428) | 6 month (n=422) | | Baseline (n=92) | 1 month (n=102) | 3 month (n=98) | 6 month (n=88) | | |
| Moderate severe | 40 (6.1) | – | 43 (8.0) | 34 (6.5) | | 32 (6.0) | – | 36 (6.8) | 25 (5.9) | | 6 (6.5) | – | 5 (5.4) | 8 (9.0) | | 20 (6.5) |
| Severe | 25 (3.8) | – | 18 (3.3) | 15 (2.9) | | 20 (3.8) | – | 16 (30) | 10 (2.4) | | 4 (4.3) | – | 2 (2.2) | 4 (4.5) | | 16 (5.2) |
| Resilience | 5.2±1.9 | – | 5.7±1.7 | 5.8±1.6 | <0.001 | 5.2±1.9 | – | 5.7±1.7 | 5.7±1.7 | <0.001 | 5.0±2.1 | – | 5.7±1.7 | 6.0±1.6 | <0.001 | 5.7±1.2 |

[a]The Chi-squared test was used for categorical variables and $t$ test was used for continuous variables.

[b]Chemsex, and commercial sex were classified as "yes" if participants reported engaging in these behaviors at least once across the two follow-up assessments. If participants reported having multiple same-sex partners at least once during the two follow-ups, they were classified as "> 1". If participants reported Inconsistent condom use at least once during the two follow-ups, they were classified as "Inconsistent". If participants reported their sexual role as receptive or versatile at least once during the two follow-ups, they were classified as "receptive or versatile". Self-efficacy for being adherent to PrEP, resilience, depressive symptoms, and PrEP-related stigma in the last column were assessed using the mean scores from the two follow-up assessments.

[c]For chemsex, condom use, sexual partner, sexual role, and commercial sex, the total sample for these variables was those who have reported being sexually active for the past study duration.

[d]PrEP-related knowledge was assessed by asking participants to identify correct dosing instructions based on their regimen (daily or ED). Only those who selected all correct options were coded as having correct PrEP adherence knowledge.

ED, event-driven; PrEP, Pre-exposure prophylaxis

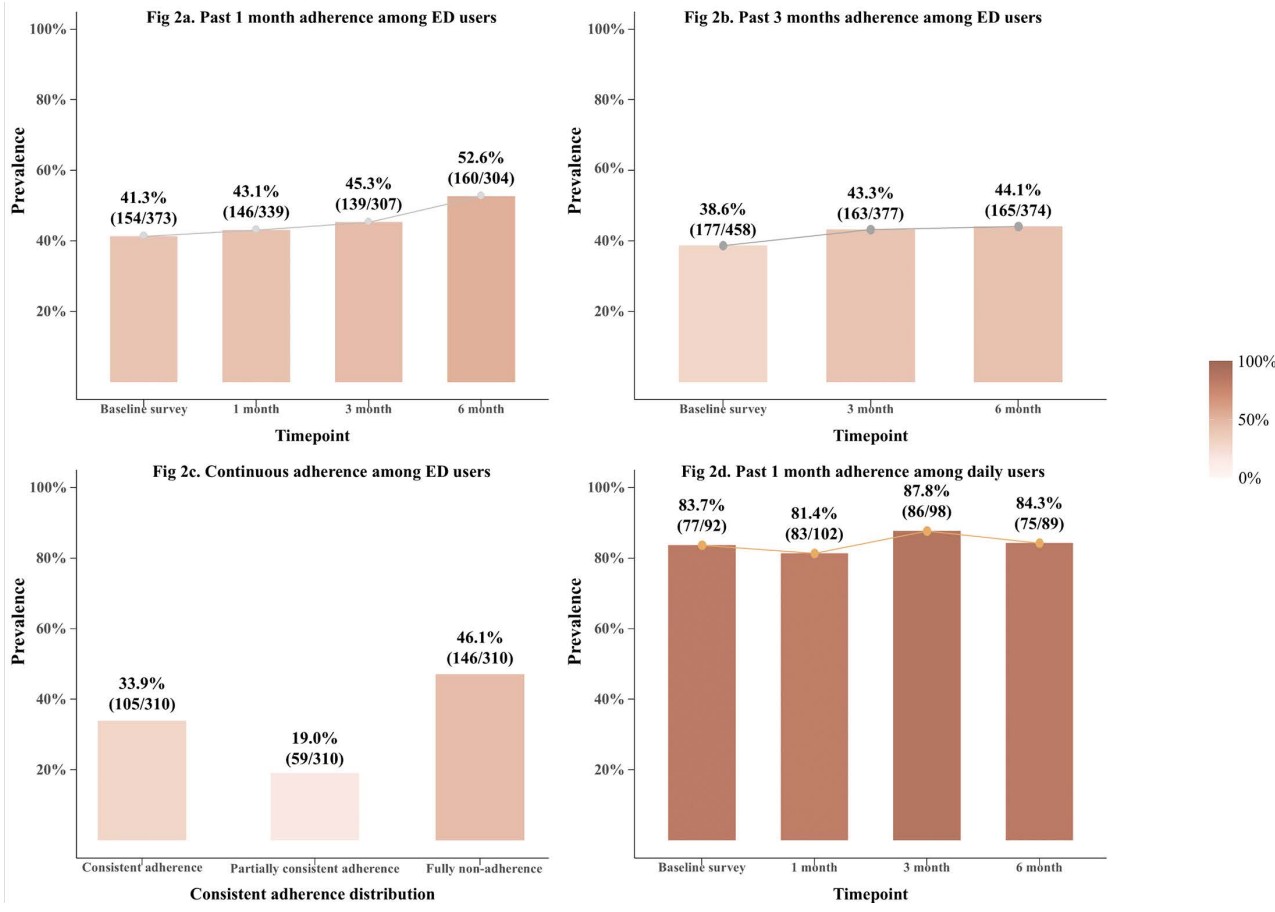

**Fig 2. Prevalence of optimal PrEP Adherence by PrEP regimens and outcome definitions in the RECOPE study.** Panel **(a)** shows the prevalence of optimal adherence in the past 1 month among event-driven (ED) PrEP users at baseline and 1-, 3-, and 6-month follow-ups. Panel **(b)** shows the prevalence of optimal adherence in the past 3 months among ED users at baseline and 3- and 6-month follow-ups. Panel **(c)** presents the distribution of consistent adherence status among ED users across the study period. Panel **(d)** shows the prevalence of optimal adherence in the past 1 month among daily PrEP users at baseline and 1-, 3-, and 6-month follow-ups.

52.6%(160/304) at the baseline, 1-, 3-, and 6-month follow-ups, respectively (Fig 2a); the prevalence of optimal adherence in the past three months was 38.6% (177/458), 43.3% (163/377), and 44.1% (165/374) at the baseline, 3-, and 6-months follow-ups, respectively (Fig 2b). Regarding the consistent adherence outcome, 33.9% (105/310) of the analyzed participants reported consistent adherence, 19.0% (59/310) reported partially consistent adherence (being adherent in one of the two follow-ups), and 46.1% (146/310) reported fully non-adherence (Fig 2c). In addition, we further examined the patterns of missing doses among non-adherent ED users in the past month at each survey wave (S3 Table). Approximately 35.2%–59.0% reported missing the two pills at time point A across the 4 waves, 38.9%–63.2% failed to take the pill at time point B, and 45.7%–64.6% missed the pill at time point C.

Among the daily regimen users, the prevalence of optimal adherence (and the median [IQR] number of pills missed in the past month) was 83.7% (2 [1.5]), 81.4% (2 [2.0]), 87.8% (2 [1.0]), and 84.3% (2 [2.0]) in the four survey waves, respectively (Fig 2d). The mean adherence rates in the past one month were 98.7% (SD = 4.3), 97.8% (SD = 10.2), 99.3% (SD = 2.4), and 97.1% (SD = 13.1) at baseline and 1-, 3-, and 6-month follow-ups, respectively (S4 Table).

Regarding the reasons for non-adherence, the most frequently cited reasons by ED users were "I used other protection (e.g., condoms)," "I forgot to take PrEP," and "I trust my partner" (S1 Fig). The most common reasons among daily users

were "I forgot to take PrEP", "I don't have access to PrEP in a timely manner", and "It was inconvenient for me to take PrEP" (S2 Fig).

## Factors associated with PrEP adherence

In Model 1 for the past 3-month adherence outcome (Table 3), greater self-efficacy (adjusted odds ratio [aOR] = 1.51, 95% confidence interval [CI] [1.31, 1.75], $p < 0.001$), taking receptive/ versatile sexual role (aOR = 1.56, 95% CI [1.01, 2.41], $p = 0.044$), having multiple same-sex partners (aOR = 0.50, 95% CI [0.28, 0.88], $p = 0.015$), and PrEP-related stigma (aOR = 0.96, 95% CI: [0.93, 0.99], $p = 0.039$) were significantly associated with optimal adherence. In Model 2 for the past 1-month adherence outcome (S5 Table), greater self-efficacy and not having multiple same-sex partners in the past three months were significantly and positively associated with PrEP adherence. In Model 3 for the consistent adherence outcome (S6 Table), older age, greater self-efficacy, and not engaging in chemsex were significantly positive associated factors.

In the univariable regression analyses using adherence rate as a continuous outcome for daily users (S7 Table), no significant associations were observed for most exposure variables, except for marital status ($b = -3.51$, 95% CI [−6.30, −0.72], $p = 0.014$) and self-efficacy ($b = 0.64$, 95% CI [0.05, 1.23], $p = 0.034$).

## Discussion

The RECOPE is among the earliest prospective cohort research targeting real-world e-consumers of PrEP in China. The study revealed that the prevalence of optimal PrEP adherence among ED regimen users, who represented the majority of real-world users in China, was low, while that among daily users was high. It also identified various prognostic

**Table 3. Factors associated with optimal PrEP adherence in the past three months based on baseline and 3- and 6-month follow-up data among ED users.**

| Variables | Model 1 on optimal adherence in the past three months (Univariate analysis) | | | Model 1 on optimal adherence in the past three months (Multivariable analysis) | | |
|---|---|---|---|---|---|---|
| | OR | 95% CI | P | aOR | 95% CI | P |
| Age | 1.01 | (0.98, 1.03) | 0.728 | 1.02 | (0.98, 1.06) | 0.239 |
| Ethnicity (Others vs. Han) | 1.42 | (0.99, 2.03) | 0.860 | 1.01 | (0.41, 2.49) | 0.984 |
| Monthly income (>3,000 CNY vs. ≤3,000 CNY) | 0.89 | (0.54, 1.46) | 0.317 | 0.67 | (0.41, 2.49) | 0.217 |
| Marital status (Unmarried/Divorced/Separated/Widowed vs. Married/Living with a Partner) | 0.67 | (0.35, 1.27) | 0.157 | 0.75 | (0.38, 1.46) | 0.386 |
| Education (College and above vs. High school or below) | 0.71 | (0.49, 1.01) | 0.685 | 1.19 | (0.42, 3.39) | 0.999 |
| Employment status (Employed vs. Students/Unemployed) | 0.95 | (0.48, 1.88) | 0.847 | 1.04 | (0.56, 1.92) | 0.889 |
| Knowledge of event-driven regimen (Correct vs. Incorrect) | 1.44 | (0.85, 2.44) | 0.206 | 1.62 | (0.98, 2.69) | 0.055 |
| Having multiple same-sex partners in the past three months (Multiple vs. One same-sex partner) | 0.35 | (0.22, 0.56) | **<0.001** | 0.50 | (0.28, 0.88) | **0.015** |
| Sexual role in the past three months (Receptive or versatile vs. Insertive) | 1.39 | (0.85, 2.26) | 0.145 | 1.56 | (1.01, 2.41) | **0.044** |
| Chemsex in the past three months (Having chemsex vs. No chemsex) | 0.40 | (0.25, 0.64) | **<0.001** | 0.72 | (0.46, 1.12) | 0.130 |
| Condom use in the past three months (Inconsistent vs. Consistent use) | 0.92 | (0.59, 1.43) | 0.829 | 0.96 | (0.61, 1.49) | 0.774 |
| Commercial sex in the past three months (Having commercial sex vs. No commercial sex) | 0.56 | (0.24, 1.26) | 0.210 | 1.73 | (0.81, 3.70) | 0.163 |
| Self-efficacy of being adherent to PrEP | 1.88 | (1.65, 2.15) | **<0.001** | 1.51 | (1.31, 1.75) | **<0.001** |
| Resilience | 1.02 | (0.91, 1.14) | 0.775 | 0.95 | (0.84, 1.07) | 0.272 |
| Depressive symptoms | 0.97 | (0.93, 1.01) | 0.140 | 0.97 | (0.93, 1.02) | 0.256 |
| PrEP-related stigma | 0.98 | (0.95, 1.01) | 0.127 | 0.96 | (0.93, 0.99) | **0.039** |

ED, event-driven; PrEP, Pre-exposure prophylaxis; CNY, Chinese Yuan; OR, odds ratio; CI, confidence interval; aOR, adjusted odds ratio.

factors potentially affecting adherence behaviors in this group, which shed light on future interventions. The study underscores the urgent need for timely surveillance and targeted interventions to improve PrEP adherence among real-world e-consumers in China.

In this study, the prevalence of optimal adherence among ED users was mostly lower than 50% across the four survey waves. Although no prior real-world study are available for direct comparison, PrEP adherence data from previous clinical/implementation/promotional trials can be useful references. For example, a non-randomized trial in China involving 503 ED users who received PrEP for free in clinics reported that 57.4%–77.8% of self-reported days involving anal intercourse were covered by appropriately timed PrEP use [29]. A study in France, which included 1,509 ED users who received PrEP for free through offline venues, indicated that 78.4% of the reported sexual acts were covered by PrEP use [30]. Both studies showed a higher prevalence of optimal adherence than the current study.

Some possible explanations for the suboptimal adherence among ED users in the RECOPE study include potential gaps in instruction and counseling on medication use among e-consumers (indicated by the findings that about one quarter of the ED users did not correctly answer all the knowledge questions), use of other promising HIV prevention strategies (indicated by the finding that the top self-reported reasons for non-adherence included 'I used other HIV prevention'), the complexity of the ED regimen, lack of a systematic reminder mechanism, and difficulty in predicting sexual activity. The findings imply that future interventions on PrEP adherence should address these important issues. To be noted, the most commonly reported reason for non-adherence among ED users in the study was the use of other prevention strategies such as condoms, which implies that some non-adherent ED users might still be able to protect themselves from HIV infection. Condoms, when used correctly and consistently, are safe and highly effective in preventing transmission of most sexually transmitted infections, including HIV [31], and can be considered as a reasonable substitute for PrEP. Future studies may consider calculating prevention-effective adherence in which PrEP use is aligned with potential risk exposure [32]. Meanwhile, the finding indicates that in order to fully protect people from HIV, it is also important to integrate health education of other HIV prevention strategies (e.g., correct and consistent use of condom, partner viral suppression [U=U]) into future PrEP services and interventions.

The prevalence of optimal PrEP adherence among the daily users in the study was above 80%, higher than those reported in previous clinical/implementation/promotional trials conducted among MSM in China (72.1%−75.1%), Brazil (74%), and West Africa (71%), even though the definition of optimal adherence in the present study was more strict than that in previous studies [29,33,34]. A possible reason is that in these previous trials, participants obtained PrEP free of charge, whereas in the present study, the participants were real-world clients who perceived themselves as individuals at high risk of HIV and bought the medication out-of-pocket. As a result, they may take medication adherence more seriously. Another possible reason for the high adherence among the daily PrEP users is the social desirability bias introduced by self-administered survey questionnaires. Despite the encouraging finding, continued adherence monitoring and interventions to support sustained optimal PrEP adherence remain essential for real-world daily PrEP users.

Self-efficacy was consistently a significant positive predictor of PrEP adherence among both ED and daily users across all the models using different outcome definitions in the study, aligning with previous research findings [35]. As a core construct in classic health behavior theories, such as the Social Cognitive Theory and Health Belief Model, self-efficacy reflects one's confidence in their ability to perform designated behaviors (e.g., following prescribed dosing) and is formed by mastery experience, vicarious experience, social persuasion, and regulation of physiological and emotional states [36]. Future PrEP adherence interventions are suggested to incorporate self-efficacy enhancement strategies (e.g., reminder system, motivational interviewing, cognitive-behavioral therapy) to promote long-term adherence and optimize the benefits of PrEP [37].

In the study, high-risk sexual behaviors were negatively associated with optimal adherence among ED users. Participants with multiple male sexual partners were more likely to report suboptimal adherence, potentially due to the unpredictability of their sexual encounters and lower HIV risk perception [14,38]. Similarly, chemsex was linked to poor adherence, which may be driven by cognitive impairment and behavioral disinhibition that interfere with timely dosing [39]. These

alarming findings underscore the importance of identifying and prioritizing PrEP users who engage in high-risk sexual behaviors but remain non-adherent, and of developing comprehensive intervention strategies (e.g., risk communication, individualized counseling, and peer-led strategies). Notably, participants who identified their sexual role as receptive or versatile, a subgroup facing higher HIV risk, were more likely to report optimal adherence in the study. It may be caused by heightened perceptions of HIV susceptibility associated with receptive anal intercourse [40].

A higher level of PrEP-related stigma was significantly and negatively associated with optimal PrEP adherence among ED users, consistent with previous studies conducted among various populations [41,42]. Stigma may discourage PrEP adherence by eliciting social undesirability, shame and guilt, internalized negative beliefs, and fear of being perceived as engaging in high-risk behavior. These effects may be amplified in MSM populations, aligning with the Minority Stress Theory, which posits that stigma-related stress undermines health-promoting behaviors [43]. Future PrEP services, both online and offline, should provide non-judgmental and affirming care that actively reduces users' perceived stigma towards PrEP, ensuring that individuals feel supported and empowered throughout their PrEP journey.

Older ED PrEP users showed a higher prevalence of consistent adherence in this study, consistent with findings from a study in the U.S. [44]. Younger individuals may face greater challenges in maintaining adherence due to limited health management skills, lower risk perception, and structural barriers such as financial or healthcare access constraints [45]. Tailored strategies for younger users may include digital reminders, HIV-related health education, and peer-based support. In the daily PrEP group, being married/living with a partner was significantly associated with higher adherence rate, consistent with findings from previous PrEP adherence research [46,47].

Several limitations should be noted in the study. First, the convenience sampling method used in the study might introduce selection bias. The respondents of the study might be more enthusiastic about PrEP and HIV prevention and thus had better medication adherence than the non-respondents. Therefore, the PrEP adherence level reported in this study might overestimate the adherence of all real-world users who bought PrEP online in China. But as no information was available for the non-respondents, the exact direction and magnitude of selection bias could not be gauged. Second, the reliance on self-reported measures to assess PrEP adherence and other socio-demographic and psycho-behavioral variables may introduce recall and social desirability biases. To minimize these biases, we rigorously designed the questionnaire, carefully delivered informed consent, conducted anonymous surveys, and performed quality checks. Nevertheless, the self-reported measures may still lead to an overestimation of PrEP adherence in the study. Future studies using more objective measurements (e.g., smart pill box, ecological momentary assessment of PrEP adherence via digital health platform) are warranted. Third, the follow-up itself may have an interventional effect on psycho-behavior variables. For example, the adherence outcome, PrEP knowledge, and self-efficacy of being adherent showed an increase to some extent from baseline to subsequent follow-ups in the study. Possible explanations include that repeated follow-up surveys may have reinforced adherence behaviors, encouraged participants to acquire PrEP-related knowledge, and enhanced their self-efficacy. This follow-up effect should be considered when interpreting the results. Fourth, the study assumed that the current online services of PrEP make it difficult for PrEP users to receive comprehensive instructions and professional counseling, which may pose distinct challenges for helping them maintain optimal medication adherence. However, the study did not include measurements that directly evaluate the participants' actual reception of online counseling and their perceived quality and satisfaction of the counseling, as well as their perceived influence or importance of counseling on adherence. Future studies are suggested to address this issue. Fifth, some other factors potentially associated with PrEP adherence (e.g., risk perception, peer support, prior health service use) were not included in the study and need further investigation. Furthermore, the convenience sampling from a single e-commerce platform may also lead to limited representativeness of the study sample. Caution is warranted when generalizing the study findings to all e-consumers of PrEP in China. For example, all the participants in the study were MSM. Although it is true that MSM account for the majority of PrEP users in the country, those with a heterosexual orientation were not captured in the study. In addition, 95% of the participants had an education level of college or above, higher than that in previous PrEP studies in China (e.g.,

70.0%−84.1% received college or above education) [48,49]. The study also reported a higher monthly income than previous studies among Chinese MSM [48,49]. Some other characteristics of the study sample were comparable to previous PrEP-related studies in China or the context of the overall Chinese population. For example, 93.6% of the study participants were Han people. According to the latest National Population Census in China, Han ethnicity accounted for 91.11% of the total population [50]. Last but not least, the study only included people who bought PrEP online. However, other PrEP delivery systems such as hospitals, clinics, and CDC are also important venues for promoting PrEP use. Future research on these dispensing venues will be needed.

The RECOPE study revealed that the medication adherence of ED PrEP users was suboptimal, while that of the daily PrEP users was relatively high. Since ED users account for the majority of real-world e-consumers of PrEP in China, the low medication adherence may be detrimental to the progress of HIV prevention and control in the country and the globe. Future targeted interventions are suggested to enhance comprehension of medication instructions, strengthen self-efficacy, and mitigate PrEP-related stigma. Special attention should be given to PrEP users who are younger and engage in high-risk sexual behaviors.

## Supporting information

**S1 Fig. Reported reasons for PrEP non-adherence across four survey time points among ED users.** The bar chart presents the distribution of self-reported reasons for not adhering to PrEP among ED users at baseline and 1-, 3-, and 6-month follow-ups.
(PPTX)

**S2 Fig. Reported reasons for PrEP non-adherence across four survey time points among daily users.** The bar chart presents the distribution of self-reported reasons for not adhering to PrEP among daily users at baseline and 1-, 3-, and 6-month follow-ups.
(PPTX)

**S1 Table. Patterns of missing survey responses.** This table presents the patterns of missing survey responses across study timepoints, showing the number and proportion of participants who missed one, two, or three surveys out of the total cohort, thereby describing participant retention in the longitudinal study.
(DOCX)

**S2 Table. Distribution of sexual behaviors, PrEP-related variables, and psychosocial variables among participants who have not initiated PrEP.** This table summarizes the distribution of demographic, sexual behavior, PrEP-related, and psychosocial variables among participants who had not initiated PrEP across four time points (baseline and 1-, 3-, and 6-month follow-ups), illustrating changes or stability in participant characteristics over the follow-up period.
(DOCX)

**S3 Table. Patterns of missing doses among non-adherent event-driven (ED) PrEP users in the past month at each survey wave.** The table presents the proportions of non-adherent ED users who missed the correct dose at each critical dosing time point of the '2-1-1' dosing schedule during the past month, across four survey waves.
(DOCX)

**S4 Table. Adherence among PrEP users across four study timepoints.** This table presents PrEP adherence across four study timepoints among users of daily and event-driven (ED) regimens. It reports the number and proportion of participants achieving optimal adherence, as well as the mean adherence rate (mean ± SD) for the daily regimen, illustrating adherence patterns and consistency over time.
(DOCX)

**S5 Table. Factors associated with optimal medication adherence in the past month across four surveys among ED users.** The table presents univariate and multivariable logistic regression results (Model 2) assessing factors associated with optimal PrEP adherence in the past one month among ED users across four survey waves. Odds ratios (OR), adjusted odds ratios (aOR), and corresponding 95% confidence intervals (CI) are reported.
(DOCX)

**S6 Table. Factors associated with consistent medication adherence during 3- and 6-month follow-ups among ED users.** This table presents univariate and multivariable logistic regression results (Model 3) examining factors associated with consistent adherence across the 3- and 6-month follow-ups. Odds ratios (OR), adjusted odds ratios (aOR), and their corresponding 95% confidence intervals (CI) are reported.
(DOCX)

**S7 Table. Univariable linear mixed-effects regression results for adherence rate (%) among daily PrEP users.** This table presents the results of univariable linear mixed-effects regression analyses examining factors associated with adherence rate (%) among daily PrEP users. It reports regression coefficients (β), 95% confidence intervals (CI), and *P*-values for demographic, behavioral, and psychosocial variables to identify potential predictors of adherence.
(DOCX)

**S1 File. Study protocol and statistical analysis plan.** This file provides details of the study design, data collection procedures, outcome definitions, and prespecified analytical methods.
(DOCX)

**S2 File. Baseline questionnaire.** This file shows the survey tools for data collection, including items on demographic characteristics, sexual behaviors, psychosocial measures, and PrEP-related variables.
(DOCX)

**S1 Checklist. STROBE checklist.** This file summarizes how the study adheres to the Strengthening the Reporting of Observational Studies in Epidemiology (STROBE) reporting guidelines. This checklist is based on the STROBE Statement—Checklist of items that should be included in reports of cohort studies (von Elm and colleagues, PLoS Medicine, 2007). The STROBE checklist is distributed under the Creative Commons Attribution License (CC BY 4.0): https://creativecommons.org/licenses/by/4.0/. More information is available at https://www.strobe-statement.org/.
(DOCX)

**S2 Checklist. CROSS checklist.** This file summarizes how the study adheres to the Consensus-Based Checklist for Reporting of Survey Studies (CROSS) reporting guidelines.
(DOCX)

## Acknowledgments

We deeply appreciate the time and effort of all the participants, recruiters, and coordinators.

## Author contributions

**Conceptualization:** Qingyu Li, Huachun Zou, Chunqing Lin, Jie Xu, Houlin Tang, Hui Xue, Sitong Luo.

**Data curation:** Qingyu Li, Jingtao Zhou, Yuhang Zhang, Sitong Luo.

**Formal analysis:** Qingyu Li, Jingtao Zhou, Yutong Xu, Min Zhang.

**Funding acquisition:** Sitong Luo.

**Investigation:** Qingyu Li, Jingtao Zhou, Yutong Xu, Jiayin Zheng, Yuhang Zhang, Siwen Huang, Zhiyi Zhao, Chi Ruan, Sitong Luo.

**Methodology:** Qingyu Li, Huachun Zou, Chunqing Lin, Min Zhang, Jie Xu, Houlin Tang, Sitong Luo.

**Project administration:** Sitong Luo.

**Resources:** Jingtao Zhou, Jiayin Zheng, Hui Xue, Sitong Luo.

**Software:** Jiaqi Cheng.

**Supervision:** Sitong Luo.

**Validation:** Jingtao Zhou, Min Zhang, Jiaqi Cheng.

**Visualization:** Zhiyi Zhao.

**Writing – original draft:** Qingyu Li.

**Writing – review & editing:** Qingyu Li.

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
