## [Editor Report · Decision Letter 0]

5 Aug 2025

Dear Dr luo,

Thank you for submitting your manuscript entitled "Medication adherence and its associated factors among oral PrEP users in China: Main results of the real-world e-consumer cohort of pre-exposure prophylaxis (RECOPE)" for consideration by PLOS Medicine.

Your manuscript has now been evaluated by the PLOS Medicine editorial staff as well as by an academic editor with relevant expertise and I am writing to let you know that we would like to send your submission out for external peer review.

In view of comments we received from the academic editor, we ask that you provide a more detailed description of the methodology, discuss whether lack of adherence accounted for sex with primary partners (i.e. clarify the definition of adherence), and that you cite related papers on telemedicine approaches for PrEP delivery, as have been conducted in the United States, for example. Additional information on the e-consumer model is needed for external review, and we also ask that you discuss further the potential bias associated with convenience sampling.

Before we can send your manuscript to reviewers, we need you to complete your submission by providing the metadata that is required for full assessment. To this end, please login to Editorial Manager where you will find the paper in the 'Submissions Needing Revisions' folder on your homepage. Please click 'Revise Submission' from the Action Links and complete all additional questions in the submission questionnaire.

For clinical studies, please upload a copy of your trial study protocol as a supporting information file. The study protocol should be the version submitted for approval to the institutional review board or ethics committee, should include any amendments to the study protocol, as well as the date of their approval by the institutional review or ethics committee. Please also detail any deviations from the study protocol in the Methods section of your manuscript. The editors will consider the protocol and study conduct prior to a final decision for external review.

Please re-submit your manuscript within two working days, i.e. by Aug 07 2025 11:59PM.

Kind regards,

Alison Farrell, Ph.D.

Senior Editor

PLOS Medicine

---

## [Decision Letter · Decision Letter 1]

23 Sep 2025

Dear Dr luo,

Many thanks for submitting your manuscript "Medication adherence and its associated factors among oral PrEP users in China: Main results of the real-world e-consumer cohort of pre-exposure prophylaxis (RECOPE)" (PMEDICINE-D-25-02683R1) to PLOS Medicine. The paper has been reviewed by subject experts and a statistician; their comments are included below and can also be accessed here: [LINK]

As you will see, the reviewers find the work of interest but have raised numerous concerns that would need to be addressed to consider a revised manuscript. In particular, the reviewers find that additional information on the study design and conduct, the representativeness of the participants, the effects of the study design on adherence, and how the data were handled and the statistical tests reported, is required, among other issues. The reviewers further find that the conclusions need to be tempered and the discussion of study limitations expanded to address the concerns raised in review. After discussing the paper with the editorial team and an academic editor with relevant expertise, I'm pleased to invite you to revise the paper in response to the reviewers' comments. We plan to send the revised paper to some or all of the original reviewers, and we cannot provide any guarantees at this stage regarding publication.

We note that an author is an employee of BlueHealth. Please amend your funder statement accordingly, as well as your competing interests statement.

Please include a protocol and statistical analysis plan, if the latter is available. Please explain what is meant by the 'main results' of RECOPE. Please include a completed STROBE checklist and update the manuscript accordingly. Please also include the exact dates of participant enrollment. Please include the survey questions as supplementary information.

Please report your survey response rates according to AAPOR recommendations https://aapor.org/standards-and-ethics/best-practices/

Please define how the population surveyed was sampled.

Please compare characteristics of respondents and nonrespondents if possible.

Were sequential waves of the survey sent? If yes, did the characteristics of respondents change over time or remain constant?

Please include the survey response rate in your Abstract.

We ask that you submit your revision by Oct 14 2025 11:59PM. However, if this deadline is not feasible, please contact me by email, and we can discuss a suitable alternative.

Don't hesitate to contact me directly with any questions (afarrell@plos.org).

Best regards,

Alison

Alison Farrell, Ph.D.

Senior Editor

PLOS Medicine

afarrell@plos.org

Comments from the academic editor:

1. The issue of representativeness of the sample. The current draft does discuss this in the limitations section but I would argue that it should have its own section or paragraph in the Discussion. The reviewers point out sampling concerns (e.g. highly educated Han participants only) beyond just the convenience sampling that could also be considered within the context of China's overall population, since most readers will not be able to judge how generalizable this study is without being informed.

2. The point made by one of the reviewers about whether trial follow-up affected outcomes. It is always a key concern for trials with behavior (adherence or retention) as an endpoint. This should be addressed.

Comments from the reviewers:

Reviewer #1: This is an interesting and timely study that evaluates PrEP adherence delivered through an "e-commerce" platform. I have reviewed Revision 1, in which the authors addressed some prior reviewer comments. However, several areas still require clarification, particularly in the Methods and Analysis sections.

1. Methodology - Additional detail needed

While telemedicine is a common approach for PrEP delivery, its implementation varies widely. Providing more detail about BluedHealth and its telehealth model would strengthen the manuscript.

1.1) General Information about BluedHealth

Please include:

- The format of the service (app or website)

- Year of establishment

- Other services offered (beyond PrEP and PEP, if applicable)

- Approximate user base or annual client volume

If this information is available in a prior publication, please cite it. I attempted to review reference 12, but it was not available in English.

1.2) Initial Telehealth Encounter and Follow-Up

- Lines 88-91 raise concerns about the telehealth approach: "However, the virtual nature of online ventures makes providing professional medication instructions and counseling difficult, which poses distinct challenges for helping consumers maintain optimal medication adherence."

Please elaborate on this point. Why would counseling differ from in-person visits, given that it does not require physical contact? If this was a concern from the outset, the study design should have included measures to evaluate this hypothesis. Additionally, none of the adherence barrier questions appear to address this issue.

1.3) Please describe the cost of PrEP in China and through this platform. Is it more affordable than conventional options?

1.4) The incentive method could influence follow-up rates. Please explain the BluedHealth voucher in more detail. What can it be used for?

1.5) It appeared that the participants were contacted outside BluedHealth during the study. Were they required to continue follow-up via telehealth in Bluedhealth? Follow-up visit rates could significantly impact adherence.

2. Results Section

2.1) For participants lost to follow-up, did they continue filling prescriptions or attending visits? (Similar concern to 1.5)

2.2) What is the rationale for using 7,000 CNY as the income threshold?

2.3) The study used PHQ-9 for depression screening. Given the range from mild to severe depression, consider classifying participants by severity rather than a binary yes/no.

2.4) Figures and Data Presentation

- Figure 2 has poor resolution and adds limited value. Consider revising, for example by combining graphs (e.g., comparing event-driven and daily PrEP in the same figure).

- PrEP adherence often varies within individuals over time. Consider presenting this variability using a heatmap to illustrate adherence patterns.

3. Discussion

3.1) The discussion should be consolidated and organized by theme. Currently, similar points are repeated in different sections/paragraphs.

3.2) The first paragraph aligns with the introduction, but subsequent sections lack coherence. For example, while 94% of participants were already using PrEP at baseline and most answered PrEP-related questions correctly, what evidence supports the statement: "inadequate instruction and counseling on medication use among e-consumers" (line 340)?

Minor Comments

- Please confirm the age range in Table 1. "28-29" appears to be an error; likely "18-29."

- Ensure consistent citation formatting. Some references are difficult to track. If a reference is only available in Chinese, please indicate this clearly.

Reviewer #2: Thanks for the opportunity to read your manuscript. My role is statistical reviewer, so I have focused on the design, data, and analysis that are presented. I have put general comments first, followed by questions relevant to a specific section of the manuscript (with a page/line reference).

This study estimates adherence to PrEP in MSM from China who sourced PreP from an electronic platform. Data is a from a prospective cohort of consumers, recruited in 2023 and 2024 from online platform for PrEP, and with follow-up data collection at 1, 3, and 6 months. The recruitment rate for the study was good (657/680 eligible participants) and a response rate at 6 months of 80% was achieved. Adherence was defined as reporting taking PrEP before each sexual encounter inline with '2-1-1' dosing, or taking daily PrEP. A range of personal characteristics and psycho-social variables were collected at baseline, as well as sexual behaviour and knowledge, stigma, and self-efficacy around PrEP use. Prevalence of adherence was reported stratified by type of PrEP use at each data collection point. A regression model estimating the association between different personal factors and adherence in the ED users, examining 1-month, 3-month, and consistent adherence.

It looks like for the regression models, an odds ratio for a model with just each factor and time was estimated, as well as an overall multivariable model with all factors included. A drawback to the multivariable model is that the interpretation of odds ratios because more difficult, because depending on the purported relationship between these, they will be a mix of 'total' effects and 'direct' effects (i.e. table 2 fallacy, https://doi.org/10.1093/aje/kws412). E.g., if age influences self-efficacy, then the odds ratio of age and self-efficacy from the multivariable model have different interpretations. Given the study aims are not causal and are more descriptive, I would consider just presenting the odds ratios from the univariate models.

P6, L116. What was the reason to include the design effect adjustment, and what parameters were used to estimate this design effect?

P7, L134. I would move the information about recruitment and participant flow to the results.

P12, L232. Was an interaction between time-point and factors included? Or was the effect of the covariate assumed to be constant over the study?

P12. L233. What covariates were included in these models?

Reviewer #3: This paper describes self-reported adherence to PrEP among a convenience sample of people obtaining PrEP online in China. Optimal (i.e., complete) adherence to event-driven PrEP was suboptimal (41-53%) and better (84-88%) for daily PrEP. Key factors identified included age, self-efficacy, and types of sexual behavior. Strengths of the paper include the real-world context, repeated measures, and sample size. Limitations include the reliance on subjective information, lack of representative sampling, and lack of attention to longitudinal prevention needs. Overall, the paper is of interest and adds to the literature on "real-world" adherence to PrEP; however, the take-home messages about challenges in PrEP adherence are well-established.

Some overarching questions:

1. The study included only MSM despite MSM not being an inclusion criterion. This point is worth noting in the paper. The sample was also highly educated and Han. If online dispensing is the primary means for getting PrEP, other delivery systems in China are needed.

2. If participants were using condoms (a primary cited reason for not taking PrEP), is PrEP adherence needed? Condoms are an effective HIV prevention tool. I recommend the authors consider the concept of prevention-effective adherence, in which PrEP use is aligned with potential risk exposure (see Haberer et al, AIDS 2015).

3. While the article is largely clear and well-written, the grammar should be reviewed by a native English speaker.

Some detailed comments follow:

Abstract

4. Line 7: I would remove "first ever", which sounds sensational (in the Abstract and in the paper).

5. Line 9: Add "modifying" after "potential".

6. Results: I would report the adherence rates for daily PrEP as well.

Introduction

7. Line 66: Adherence was a key issue in the clinical trials as well (e.g., the low adherence in FEM-PrEP and VOICE showed no efficacy; high documented adherence in Partners PrEP was a key factor in the approval and normative recommendations of PrEP). I would not over-emphasize its importance in the real-world (vs trials).

8. Line 69: While percent adherence certainly has a role in its effectiveness, it is also important to align PrEP use with potential exposure (see my comment above). Risk is not constant and may explain how seemingly low PrEP adherence can still be effective in HIV prevention.

9. Line 90: I wonder why being online makes medication instructions and counseling difficult. The authors themselves argue it is more convenient and less stigmatized.

Methods

10. Line 108: Please indicate the corresponding author's affiliated organization directly.

11. Line 118: I don't know that the equation for a sample size calculation is necessary for this type of publication (i.e., the described methods and reference seem sufficient).

12. Lines 125-127: What does being "qualified for PrEP use" mean? How did the authors establish whether someone had "communication barriers or reading disabilities"? It seems that not completing the baseline survey was also considered as an exclusion criterion (line 137), which should be more clearly explained with the other exclusion criteria.

13. Line 132: I understand the consenting procedures, but "oral" does not seem to be the appropriate descriptor (perhaps "anonymous consent"?).

14. Line 142 and 149-150: I would expect to see the follow-up rates/sample sizes in the Results section. I am also confused how the follow-up rate was 90.1% at baseline when completion of the survey was a requirement for enrollment.

15. Line 168: It appears the baseline adherence depended on 1 month of PrEP use, but the enrollment criteria did not specify that requirement. Did all participants have at least 1 month of PrEP use prior to enrollment?

16. Line 173: The tone of the question, "Have you ever failed to take PrEP on time and in a correct dosage?" sounds judgmental. I suspect there may be substantial social desirability in the self-report (for both daily and ED dosing), which may need more discussion than a brief mention in the limitations section.

17. Line 177: Was a list of reasons given for possible non-adherence, or were the questions open-ended?

18. Lines 218 and 221: Do the Cronbach alphas refer to the data in this study or from a validation cohort (e.g., in the literature)?

19. Line 229: It seems like the sample size was large enough for at least univariable regression among daily users. The authors may wish to present that data.

20. Were all variable used in the multivariable models? Please clarify and if some were excluded, provide selection process.

Results

21. Figure 1 should separate out the reasons for ineligibility and refusal to participate in the study. Also, is a cascade-style of presenting participation in the cohort appropriate? For example, did any participants miss a 1-month follow-up survey but then complete a 3- or 6-month follow-up survey?

22. Table 2 was difficult to read given its size. I would reformat it to make it easier to digest. I would also present depression as none, mild, or moderate/severe. Reporting mild to severe as one category is hard to interpret.

23. I suggest the authors give the actual adherence rates for daily PrEP, not just the dichotomized data. A linear approach would also enable predictive analysis (see my comment above).

24. The authors may want to look at ED PrEP use with partner types (e.g., main vs casual) and compare the findings to similar data from elsewhere (e.g., Jongen et al JIAS 2021).

25. Table 3. I would give the actual p-values for each variable rather than categorize them as <0.05, <0.01, or <0.001.

Discussion

26. Line 355: The high adherence among daily users could be due to social desirability bias (see my comment above). If true, however, it seems the conclusion may be to expand access to daily PrEP, not further educational programs, etc (although I agree those are still needed for people not accessing PrEP or adhering well).

27. The authors state "urgent needs" at least three times. I would be selective in prioritizing truly urgent action.

28. Line 413: Can the authors explain what they mean by "intelligent monitoring approaches"?

29. As noted above, I suggest the authors comment on adherence in relation to potential variable need for PrEP use over time.

---

* Please upload any figures associated with your paper as individual TIF or EPS files with 300dpi resolution at resubmission; please read our figure guidelines for more information on our requirements: http://journals.plos.org/plosmedicine/s/figures. While revising your submission, we strongly recommend that you use PLOS's NAAS tool (https://ngplosjournals.pagemajik.ai/artanalysis) to test your figure files. NAAS can convert your figure files to the TIFF file type and meet basic requirements (such as print size, resolution), or provide you with a report on issues that do not meet our requirements and that NAAS cannot fix.

After uploading your figures to PLOS's NAAS tool - https://ngplosjournals.pagemajik.ai/artanalysis, NAAS will process the files provided and display the results in the "Uploaded Files" section of the page as the processing is complete.

If the uploaded figures meet our requirements (or NAAS is able to fix the files to meet our requirements), the figure will be marked as "fixed" above. If NAAS is unable to fix the files, a red "failed" label will appear above.

When NAAS has confirmed that the figure files meet our requirements, please download the file via the download option, and include these NAAS processed figure files when submitting your revised manuscript.

* Please revise your funding, COI and Data Availability statements. See the website for additional information (https://journals.plos.org/plosmedicine/s/competing-interests;
https://journals.plos.org/plosmedicine/s/competing-interests;
https://journals.plos.org/plosmedicine/s/disclosure-of-funding-sources).

* Please ensure that the study is reported according to the appropriate guideline and include the completed checklist as Supporting Information. When completing the checklist, please use section and paragraph numbers, rather than page numbers. Please add the following statement, or similar, to the Methods: "This study is reported as per [XXXX] guideline (S1 Checklist)."

FIGURES AND TABLES

SUPPLEMENTARY MATERIAL

REFERENCES

STUDY TYPE-SPECIFIC REQUESTS

OBSERVATIONAL STUDIES

* Abstract: Please include the study design, population and setting, number of participants, years during which the study took place (enrollment and follow up), length of follow up, and main outcome measures.

* Please ensure that the study is reported according to the STROBE (or appropriate STOBE extension) guideline (available from: https://www.equator-network.org/reporting-guidelines/strobe) and include the completed STROBE (or STROBE extension) checklist as Supporting Information. Please add the following statement, or similar, to the Methods: "This study is reported as per the Strengthening the Reporting of Observational Studies in Epidemiology (STROBE) guideline (S1 Checklist)." When completing the checklist, please use section and paragraph numbers, rather than page numbers.

* [FOR POPULATION HEALTH/REGISTRY STUDIES] Please ensure that the study is reported according to the RECORD guideline (available from https://www.record-statement.org) and include the completed checklist as Supporting Information. Please add the following statement, or similar, to the Methods: "This study is reported as per the Reporting of Studies Conducted using Observational Routinely-Collected Data (RECORD) guideline (S1 Checklist)." When completing the checklist, please use section and paragraph numbers, rather than page numbers.

* [FOR POPULATION HEALTH ESTIMATES] Please ensure that the study is reported according to the GATHER statement (available from https://www.equator-network.org/reporting-guidelines/gather-statement) and include the completed checklist as Supporting Information. Please add the following statement, or similar, to the Methods: "This study is reported as per the Guidelines for Accurate and Transparent Health Estimates Reporting (GATHER) statement (S1 Checklist)." When completing the checklist, please use section and paragraph numbers, rather than page numbers.

* [FOR MEDIATION ANALYSES] We recommend that the study is reported according to the AGReMA statement (https://agrema-statement.org/#:~:text=AGReMA%20is%20an%20evidence%2D%20and,randomised%20trials%20and%20observational%20studies) and include the completed checklist as Supporting Information. Please add the following statement, or similar, to the Methods: "This study is reported as per the Guideline for Reporting Mediation Analyses (AGReMA) statement (S1 Checklist)." When completing the checklist, please use section and paragraph numbers, rather than page numbers.

* For all observational studies, in the manuscript text, please indicate: (1) the specific hypotheses you intended to test, (2) the analytical methods by which you planned to test them, (3) the analyses you actually performed, and (4) when reported analyses differ from those that were planned, transparent explanations for differences that affect the reliability of the study's results. If a reported analysis was performed based on an interesting but unanticipated pattern in the data, please be clear that the analysis was data driven.

* Please state in the Methods section whether the study had a prospective protocol or analysis plan. If a prospective analysis plan (from your funding proposal, IRB or other ethics committee submission, study protocol, or other planning document written before analyzing the data) was used in designing the study, please include the relevant document(s) with your revised manuscript as a Supporting Information file to be published alongside your study and cite it in the Methods section. A legend for this file should be included at the end of your manuscript. If no such document exists, please make sure that the Methods section transparently describes when analyses were planned, and when/why any data-driven changes to analyses took place. Changes in the analysis, including those made in response to peer review comments, should be identified as such in the Methods section of the paper, with rationale.

SURVEY-BASED STUDIES

* Please ensure that the study is reported according to the CROSS guideline (https://www.equator-network.org/reporting-guidelines/a-consensus-based-checklist-for-reporting-of-survey-studies-cross/) and include the completed CROSS checklist as Supporting Information. Please add the following statement, or similar, to the Methods: "This study is reported as per A Consensus-Based Checklist for Reporting of Survey Studies (CROSS) guideline (S1 Checklist)." When completing the checklist, please use section and paragraph numbers, rather than page numbers.

* Please report your survey response rates according to AAPOR recommendations (https://aapor.org/standards-and-ethics/best-practices/)

* Please define how the population surveyed was sampled.

* Please compare characteristics of respondents and nonrespondents if possible.

* If sequential waves of the survey were sent, please specify whether the characteristics of respondents changed over time or remained constant.

* Please include the survey response rate in the Abstract.

* Please include a copy of the survey in the supplementary files.

QUALITATIVE STUDIES

* Please report your qualitative study according to the appropriate study design provided at (http://www.equator-network.org/?post_type=eq_guidelines&eq_guidelines_study_design=qualitative-research&eq_guidelines_clinical_specialty=0&eq_guidelines_report_section=0&s=) and provide the relevant completed checklist as a supplemental file. In the checklist, please include sufficient text excerpted from the manuscript to explain how you accomplished all applicable items. When completing checklists, please use section and paragraph numbers, rather than page numbers.

* We recommend that authors use the COREQ checklist, or other relevant checklists listed by the Equator Network, such as the SRQR, to ensure complete reporting (see: http://www.equator-network.org/?post_type=eq_guidelines&eq_guidelines_study_design=qualitative-research&eq_guidelines_clinical_specialty=0&eq_guidelines_report_section=0&s=). Please add the following statement, or similar, to the Methods: "This study is reported as per the Consolidated criteria for reporting qualitative research (COREQ): a 32-item checklist for interviews and focus groups (S1 Checklist)."

* In general, we expect qualitative studies to include the following: 1) defined objectives or research questions; 2) description of the sampling strategy, including rationale for the recruitment method, participant inclusion/exclusion criteria and the number of participants recruited; 3) detailed reporting of the data collection procedures; 4) data analysis procedures described in sufficient detail to enable replication; 5) a discussion of potential sources of bias; and 6) a discussion of limitations.

---

## [Decision Letter · Decision Letter 2]

5 Dec 2025

Dear Dr luo,

Many thanks for submitting your manuscript "Medication adherence and its associated factors among oral PrEP users in China: Main results of the real-world e-consumer cohort of pre-exposure prophylaxis (RECOPE)" (PMEDICINE-D-25-02683R2) to PLOS Medicine. The paper has been reviewed by subject experts and a statistician; their comments are included below and can also be accessed here: [LINK]

As you will see, the reviewers find the revised manuscript improved and addresses most of their concerns. The remaining comments of reviewers 2, 3 and the academic editor are appended below. Please note that we require a response and/or qualification in the text regarding the statistical reviewer's concerns, more in depth discussion of potential selection bias and of the role of other HIV prevention strategies, and their implications of these issues for your results and conclusions. Please also ensure that your conclusions reflect event driven PrEP adherence. After discussing the paper with the editorial team and an academic editor with relevant expertise, I'm pleased to invite you to revise the paper in response to the reviewers' comments. We plan to send the revised paper to some or all of the original reviewers, and we cannot provide any guarantees at this stage regarding publication.

Please include a completed CROSS checklist with your revised manuscript (https://www.equator-network.org/reporting-guidelines/a-consensus-based-checklist-for-reporting-of-survey-studies-cross/).

We ask that you submit your revision by Dec 26 2025 11:59PM. However, if this deadline is not feasible, please contact me by email, and we can discuss a suitable alternative.

Don't hesitate to contact me directly with any questions (afarrell@plos.org).

Best regards,

Alison

Alison Farrell, Ph.D.

Senior Editor

PLOS Medicine

afarrell@plos.org

Comments from the academic editor:

1. I continue to feel that the issue of sampling selection bias is under-emphasized, in particular the concern that respondents were more likely to be enthusiastic about PrEP and adherent to it than non-respondents. The statement "It is possible that those with better PrEP adherence might be more willing to participate in the study" is not sufficient, in my view. Please consider in more depth how such a bias would affect their results. If respondents represent the best adherers, then true adherence in the overall population would likely be lower than reported, not higher.

2. The authors are too dismissive of alternative prevention strategies, such as condoms. The reviewer who mentioned that there are multiple ways to prevent HIV infection and that risk has to be considered holistically was correct. The manuscript dismisses the finding that the "top self reported reasons for non-adherence included 'I used other HIV prevention'." The rest of that paragraph goes on to dismiss self-reported condom use as unreliable, while fully accepting self-reported PrEP use. This seems particularly problematic for ED PrEP, where condom use seems like a reasonable substitute for PrEP. I would suggest that the authors consider the potential trading off of prevention methods as a valid reason for not using ED Prep, rather than an example of non-adherence, and think through the implications accordingly.

3. The first sentence of the conclusion is not accurate, as adherence to daily oral PrEP was rather high. The results need to be consistently limited to ED PrEP, to avoid a very misleading takeaway message.

Comments from the reviewers:

Reviewer #2: Thanks for the revised manuscript and responses to my original review. The updates to the methods and results have clarified most of the queries from my original review.

My main outstanding issue is the interpretation of odds ratios from the multivariable model. It is reasonable to want to understand which covariates are associated with adherence without confounding, but the output (coefficients and p-values) of multivariable model with all covariates included assumes a very particular causal model where none of the covariates are mediators of other others. The approach used is more like a prognostic factor study (e.g. doi:10.1371/journal.pmed.1001380) and if this the aim, regression models this should be made clear. Otherwise, the causal model needs to be proposed to ensure that direct effects are not confused with total effects in the model output.

L324. It would be more accurate to say that 'there was no evidence of a difference (p>0.05) in the distributions of socio-demographics overall and by regimen throughout the survey waves'.

L341. I would rephrase this as "There was good evidence (p<0.05) of a difference in sexual activity status, …."

Reviewer #3: I appreciate the opportunity to re-review this paper, which is much improved. I have the following remaining concerns.

1. Abstract- When reporting percentages, I would provide the actual numbers as well. Specifically, when presenting the prevalence of optimal adherence, it is not clear which denominators were used (e.g., 680 eligible, 657 completing baseline survey, or the number responding at each time point). Alternatively, the authors could clarify the denominator in the text.

2. Fig 1 flowchart- I appreciate the revisions, but I would add a footnote to the figure indicating that some participants completed the survey at different time points and refer to the S1 Table. To this point, in the S1 Table, I would reserve use of the term "lost to follow-up" for participants who could no longer be reached in the study. I believe the authors could substitute "missing" for each data time point.

3. I respect the authors' decision not to incorporate the concept of prevention-effective adherence, and I appreciate the suggestion of considering it in future analysis. However, the rationale that use of "other HIV prevention" may not be correct seems paternalistic and inconsistent. While indeed, real-world use of condoms may not be effective, participants reporting "other HIV prevention" may have done so because of confidence in their use of condoms. They may have also been referring to U=U, which is well acknowledged as effective (i.e., the HPTN 052 study). The authors accept the accuracy of other forms of self-report (e.g., self-reported sex for determining baseline adherence... see R3.15). Additionally, the WHO recommendation for using condoms with PrEP (cited by the authors) is primarily based on the potential for other STIs.

4. R3.26 (pg 70) The high adherence to daily PrEP seems like it should be considered as a recommended for individuals who might be interested in it. At a minimum, the conclusion in the main paper that PrEP adherence was low should be qualified to low event driven adherence (as it is in the Abstract).

---

GENERAL EDITORIAL REQUESTS

* At this stage, we ask that you include a short, non-technical Author Summary of your research to make findings accessible to a wide audience that includes both scientists and non-scientists. The Author Summary should immediately follow the Abstract in your revised manuscript. This text is subject to editorial change and should be distinct from the scientific abstract. Ideally each sub-heading should contain 2-3 single sentence, concise bullet points containing the most salient points from your study. In the final bullet point of ‘What Do These Findings Mean?’ Please include the main limitations of the study in non-technical language.

Please see our author guidelines for more information: https://journals.plos.org/plosmedicine/s/revising-your-manuscript#loc-author-summary.

* Please confirm that your title complies with PLOS Medicine's style. Your title must be nondeclarative and not a question. It should begin with main concept if possible. "Effect of" should be used only if causality can be inferred, i.e., for an RCT. Please place the study design ("A randomized controlled trial," "A retrospective study," "A modelling study," etc.) in the subtitle (ie, after a colon).

* Please confirm that your abstract complies with our requirements, including format (three sections: Background, Methods and Findings, and Conclusions) and providing all the information relevant to this study type https://journals.plos.org/plosmedicine/s/submission-guidelines#loc-abstract

* Please ensure that the Introduction ends with a clear description of the study question or hypothesis.

* Please ensure that all abbreviations are defined at first use throughout the text.

* Please confirm that all numbers presented in the abstract are present and identical to numbers presented in the main manuscript text.

**Please use the active voice throughout and request the assistance of a native English speaker to correct the grammar, phrasing and spelling throughout the text.

* Please remove the 'conclusions' subheading from the discussion. Please also remove any other subheadings from the discussion.

"* Statistical reporting: Please revise throughout the manuscript, including tables and figures.

- Please report statistical information as follows to improve clarity for the reader ""22% (95% CI [13,28]; p</=)"".

- Please separate upper and lower bounds with commas instead of hyphens as the latter can be confused with reporting of negative values.

- Please repeat statistical definitions (HR, CI etc.) for each set of parentheses."

* In the abstract, please include the important dependent variables that are adjusted for in the analyses.

* For authors with ties to industry, please indicate whether any of the interests has a financial stake in the results of the current study.

* It appears that one or more study authors is affiliated with one or more of the agencies that funded the study. Thus, the statement “The funders had no role in study design, data collection and analysis, decision to publish, or preparation of the manuscript” does not apply. Please revise the Financial Disclosure accordingly, as in "[Author name] is [author's role] at [funding agency]. The funders had no other role in study design…..”

* The funding statement should include: specific grant numbers, initials of authors who received each award, URLs to sponsors’ websites. Also, please state whether any sponsors or funders (other than the named authors) played any role in study design, data collection and analysis, the decision to publish, or preparation of the manuscript. If they had no role in the research, include this sentence: “The funders had no role in study design, data collection and analysis, decision to publish, or preparation of the manuscript.”

* Please upload any figures associated with your paper as individual TIF or EPS files with 300dpi resolution at resubmission; please read our figure guidelines for more information on our requirements: http://journals.plos.org/plosmedicine/s/figures. While revising your submission, we strongly recommend that you use PLOS's NAAS tool (https://ngplosjournals.pagemajik.ai/artanalysis) to test your figure files. NAAS can convert your figure files to the TIFF file type and meet basic requirements (such as print size, resolution), or provide you with a report on issues that do not meet our requirements and that NAAS cannot fix.

After uploading your figures to PLOS's NAAS tool - https://ngplosjournals.pagemajik.ai/artanalysis, NAAS will process the files provided and display the results in the "Uploaded Files" section of the page as the processing is complete.

If the uploaded figures meet our requirements (or NAAS is able to fix the files to meet our requirements), the figure will be marked as "fixed" above. If NAAS is unable to fix the files, a red "failed" label will appear above.

When NAAS has confirmed that the figure files meet our requirements, please download the file via the download option, and include these NAAS processed figure files when submitting your revised manuscript.

* If the data are not freely available, please describe briefly the ethical, legal, or contractual restriction that prevents you from sharing it. Please also include an appropriate contact (web or email address) for inquiries (this cannot be a study author).

* Please ensure that the study is reported according to the CROSS guideline and include the completed CROSS checklist as Supporting Information. When completing the checklist, please use section and paragraph numbers, rather than page numbers. Please add the following statement, or similar, to the Methods: "This study is reported as per CROSS guideline (S1 Checklist).

Please report your survey response rates according to AAPOR recommendations https://aapor.org/standards-and-ethics/best-practices/

Please define how the population surveyed was sampled.

Please compare characteristics of respondents and nonrespondents if possible.

Were sequential waves of the survey sent? If yes, did the characteristics of respondents change over time or remain constant?

Please include the survey response rate in your Abstract.

FIGURES AND TABLES

* Please provide the unadjusted comparisons as well as the adjusted comparisons in all relevant Tables.

* Please specify the variables controlled for in all relevant Tables.

SUPPLEMENTARY MATERIAL

REFERENCES

STUDY TYPE-SPECIFIC REQUESTS -

OBSERVATIONAL STUDIES

* Abstract: Please include the study design, population and setting, number of participants, years during which the study took place (enrollment and follow up), length of follow up, and main outcome measures.

* Please ensure that the study is reported according to the STROBE (or appropriate STOBE extension) guideline (available from: https://www.equator-network.org/reporting-guidelines/strobe) and include the completed STROBE (or STROBE extension) checklist as Supporting Information. Please add the following statement, or similar, to the Methods: "This study is reported as per the Strengthening the Reporting of Observational Studies in Epidemiology (STROBE) guideline (S1 Checklist)." When completing the checklist, please use section and paragraph numbers, rather than page numbers.

* [FOR POPULATION HEALTH/REGISTRY STUDIES] Please ensure that the study is reported according to the RECORD guideline (available from https://www.record-statement.org) and include the completed checklist as Supporting Information. Please add the following statement, or similar, to the Methods: "This study is reported as per the Reporting of Studies Conducted using Observational Routinely-Collected Data (RECORD) guideline (S1 Checklist)." When completing the checklist, please use section and paragraph numbers, rather than page numbers.

* [FOR POPULATION HEALTH ESTIMATES] Please ensure that the study is reported according to the GATHER statement (available from https://www.equator-network.org/reporting-guidelines/gather-statement) and include the completed checklist as Supporting Information. Please add the following statement, or similar, to the Methods: "This study is reported as per the Guidelines for Accurate and Transparent Health Estimates Reporting (GATHER) statement (S1 Checklist)." When completing the checklist, please use section and paragraph numbers, rather than page numbers.

* [FOR MEDIATION ANALYSES] We recommend that the study is reported according to the AGReMA statement (https://agrema-statement.org/#:~:text=AGReMA%20is%20an%20evidence%2D%20and,randomised%20trials%20and%20observational%20studies) and include the completed checklist as Supporting Information. Please add the following statement, or similar, to the Methods: "This study is reported as per the Guideline for Reporting Mediation Analyses (AGReMA) statement (S1 Checklist)." When completing the checklist, please use section and paragraph numbers, rather than page numbers.

* For all observational studies, in the manuscript text, please indicate: (1) the specific hypotheses you intended to test, (2) the analytical methods by which you planned to test them, (3) the analyses you actually performed, and (4) when reported analyses differ from those that were planned, transparent explanations for differences that affect the reliability of the study's results. If a reported analysis was performed based on an interesting but unanticipated pattern in the data, please be clear that the analysis was data driven.

* Please state in the Methods section whether the study had a prospective protocol or analysis plan. If a prospective analysis plan (from your funding proposal, IRB or other ethics committee submission, study protocol, or other planning document written before analyzing the data) was used in designing the study, please include the relevant document(s) with your revised manuscript as a Supporting Information file to be published alongside your study and cite it in the Methods section. A legend for this file should be included at the end of your manuscript. If no such document exists, please make sure that the Methods section transparently describes when analyses were planned, and when/why any data-driven changes to analyses took place. Changes in the analysis, including those made in response to peer review comments, should be identified as such in the Methods section of the paper, with rationale.

SURVEY-BASED STUDIES

* Please ensure that the study is reported according to the CROSS guideline (https://www.equator-network.org/reporting-guidelines/a-consensus-based-checklist-for-reporting-of-survey-studies-cross/) and include the completed CROSS checklist as Supporting Information. Please add the following statement, or similar, to the Methods: "This study is reported as per A Consensus-Based Checklist for Reporting of Survey Studies (CROSS) guideline (S1 Checklist)." When completing the checklist, please use section and paragraph numbers, rather than page numbers.

* Please report your survey response rates according to AAPOR recommendations (https://aapor.org/standards-and-ethics/best-practices/)

* Please define how the population surveyed was sampled.

* Please compare characteristics of respondents and nonrespondents if possible.

* If sequential waves of the survey were sent, please specify whether the characteristics of respondents changed over time or remained constant.

* Please include the survey response rate in the Abstract.

* Please include a copy of the survey in the supplementary files.

---

## [Decision Letter · Decision Letter 3]

21 Jan 2026

Dear Dr. luo,

Thank you very much for re-submitting your manuscript "Medication adherence and its associated factors among oral PrEP users in China: Main results of the real-world e-consumer cohort of pre-exposure prophylaxis (RECOPE)" (PMEDICINE-D-25-02683R3) for review by PLOS Medicine.

I have discussed the paper with my colleagues and the academic editor and it was also seen again by one reviewer. I am pleased to say that provided the remaining editorial and production issues are dealt with we are planning to accept the paper for publication in the journal.

[LINK]

We look forward to receiving the revised manuscript by Jan 28 2026 11:59PM.

Sincerely,

Alison Farrell, Ph.D.

Senior Editor

PLOS Medicine

plosmedicine.org

Requests from Editors:

Title: Please revise to “Medication adherence and its associated factors among oral pre-exposure prophylaxis (PrEP) users in China: The Real-world E-consumer Cohort of PrEP (RECOPE) study”

Line 23: Revise “The cohort study” to “This cohort study”

Line 32: please rephrase or complete the sentence. “were fitted” to what?

Line 33: Please remove paragraph break.

Line 29: were participants asked to complete e-questionnaires at 1, 3 and 6 months? If yes, please so state here.

Line 35: what ‘visit’ does this refer to? Please revise. Do you mean timepoints?

Line 48: use ‘account’ instead of ‘accounted’.

Line 49: use ‘targeted’ instead of ‘target’

Line 51: replace ‘being adhere’ with ‘being adherent’ or preferably by ‘maintaining adherence’

Please move the Author Summary to after the Abstract.

Please correct the grammar in Figure 1 (first box upper right). Please keep verb tense constant (use past tense), use ‘mo’ instead of ‘mon’ as an abbreviation for month, please clarify/revise use of ‘break-off’.

Please correct grammar in Fig. 2c panel title “continuous adherence during among ED users”

Please do not use ‘visit’ for x axis titles in Figure 2 or elsewhere if what you mean is timepoint.

Line 66: replace ‘toll of HIV’ with ‘tool for HIV’

Line 77: delete “we can only locate”

Line 78: delete “which”

Line 83: revise “which is particularly accelerated” to “and whose uptake was accelerated”

Line 103: replace “needs to” with “must”

Line 108-109: suggest deleting “a poorer accessibility” and starting with ‘a more complex procedure’

Line 112-114: claim of primacy—if other studies have now been performed evaluating medication adherence among e-consumers of PrEP, please delete this sentence.

Line 115: ‘trails’ should be ‘trials’

Line 118: replace “easier access” with “greater accessibility”

Line 129: replace ‘STORBE’ with ‘STROBE’

Lines 132-145: This section reads as an advertisement for BluedHealth. Please remove any marketing material language and phrasing.

Line 312: explain 'break-off'

Line 378: should be 'non-adherent' not 'non-adhere'

Line 441: should be 'non-adherent' not 'non-adhere'

Line 452: delete 'all'

Line 473: please correct grammar. "suggested incorporate' should be "suggested to incorporate".

Line 485: remove "fortunately" as may be considered demeaning. Suggest replacing with "notably"

Line 499: delete "Additionally,"

Line 514-515: delete "were unavailable" due to redundancy. "But as no information were available " should be "But as no information was available"

Line 525: should be "being adherent"

Line 526-529: please correct grammar in this sentence.

Line 533: measurement should be plural

Line 558: should be "accounts" not "accounted"

Please reduce the repetition of ideas and concepts in the Discussion.

For each table, define the abbreviations used in the footnotes.

Please make formatting style consistent across the top of Tables 1 and 2.

Please avoid use of homosexual and use same sex instead (e.g. lines 39, 402, Table 2 footnotes, Table 3, Author summary. Please check the supplementary information and revise if necessary.).

Please check throughout for grammatical errors.

Please seek the assistance of a native English speaker when revising your manuscript.

Please do not use the term ‘visit’ for survey questionnaire timepoints in the main manuscript or the supplementary documents.

Please clarify whether written or oral informed consent was obtained and add to Methods.

* If any included images of branded products are included in the manuscript files, please confirm that you have appropriate permission or you may wish to remove these images. For more information, please see our guidelines: https://journals.plos.org/plosmedicine/s/figures#loc-licenses-and-copyright and https://journals.plos.org/plosmedicine/s/licenses-and-copyright

* Please review your text for claims of novelty or primacy (e.g. 'for the first time') and remove this language. In addition, please check that any use of statistical terms (such as trend or significant) are supported by the data, and if not please remove them.

When completing the CROSS checklist, please use section and paragraph numbers, rather than page numbers. Note that page numbers are currently used.

* Please compare characteristics of respondents and nonrespondents if possible.

* If sequential waves of the survey were sent, please specify whether the characteristics of respondents changed over time or remained constant.

Please report your survey response rates according to AAPOR recommendations https://aapor.org/standards-and-ethics/best-practices/

* Please provide titles and legends for all figures and tables (including those in Supporting Information files). Please define all acronyms used in each figure or table in its corresponding legend.

Comments from Reviewers:

Reviewer #2: The changes to the manuscript that position the multivariable regression analyses as prognosis factor research resolve my query from the last review.

[LINK]

---

## [Editor Report · Decision Letter 4]

5 Feb 2026

Dear Dr luo,

On behalf of my colleagues and the Academic Editor, Sydney Rosen, I am pleased to inform you that we have agreed to publish your manuscript "Medication adherence and its associated factors among oral pre-exposure prophylaxis (PrEP) users in China: The Real-world E-consumer Cohort of PrEP (RECOPE)" (PMEDICINE-D-25-02683R4) in PLOS Medicine.

Please also note the remaining editorial requests (line numbers are from the clean word doc):

Line 181: consent in this setting is not oral, so please delete 'an oral'.

Line 440: add 'study' after RECOPE

Please provide URLs for the funders in the metadata.

Please remove reference 12 citation on line 117 as it does not relate to the statement. Please move the citation elsewhere or replace with a relevant citation.

Line 135: replace 'were' with 'is'

Both 'counselling' and 'counseling' are used in the text. Please use one spelling only and correct accordingly.

Please revise title to "Medication adherence and its associated factors among oral pre-exposure prophylaxis (PrEP) users in China: The Real-world E-consumer Cohort of PrEP study" . We don't need the abbreviation (RECOPE) in the title.

PRESS

Sincerely,

Alison Farrell, Ph.D.

Senior Editor

PLOS Medicine